# Generalization of learned responses in the mormyrid electrosensory lobe

**Conor Dempsey[1], LF Abbott[1,2], Nathaniel B Sawtell[1]\***

[1]Department of Neuroscience, Zuckerman Mind Brain Behavior Institute, Columbia University, New York, United States; [2]Department of Physiology and Cellular Biophysics, Columbia University, New York, United States

**Abstract** Appropriate generalization of learned responses to new situations is vital for adaptive behavior. We provide a circuit-level account of generalization in the electrosensory lobe (ELL) of weakly electric mormyrid fish. Much is already known in this system about a form of learning in which motor corollary discharge signals cancel responses to the uninformative input evoked by the fish's own electric pulses. However, for this cancellation to be useful under natural circumstances, it must generalize accurately across behavioral regimes, specifically different electric pulse rates. We show that such generalization indeed occurs in ELL neurons, and develop a circuit-level model explaining how this may be achieved. The mechanism involves regularized synaptic plasticity and an approximate matching of the temporal dynamics of motor corollary discharge and electrosensory inputs. Recordings of motor corollary discharge signals in mossy fibers and granule cells provide direct evidence for such matching.

DOI: https://doi.org/10.7554/eLife.44032.001

## Introduction

A learned response that is adaptive only in the precise context in which it was learned is of limited value in the real world. Though cellular and synaptic underpinnings of learning have been elucidated in many systems, less is known about the mechanisms that allow learning to generalize appropriately to conditions different from those in which the learning originally took place (*Censor, 2013*; *Fahle, 2005*; *Poggio and Bizzi, 2004*). We address the question of generalization of learned responses in the passive electrosensory system of weakly electric mormyrid fish. These fish, like a number of other aquatic animals, possess a specialized class of electroreceptors on their skin that are sensitive to minute, low-frequency electrical fields, such as those emitted by other animals in the water (*Engelmann et al., 2010*; *Enikolopov et al., 2018*; *von der Emde and Bleckmann, 1998*). However, the detection and processing of such signals is complicated by the fact that mormyrid fish also emit their own pulsed electric fields, known as electric organ discharges (EODs). Though EODs are used for sensing nearby objects through active electrolocation as well as for communication with conspecifics (processes mediated by separate classes of electroreceptors), they also strongly activate the receptors subserving passive electrolocation, inducing a ringing pattern of activation that persists for ~200 ms (*Bell and Russell, 1978*). If left uncancelled, these responses to the fish's own EOD could impede the detection and processing of behaviorally-relevant signals such as prey (*Enikolopov et al., 2018*).

Past work has suggested that this problem is solved in ELL neurons through the integration of electrosensory input and corollary signals (CD) related to the motor command to discharge the electric organ (*Bell et al., 1981*). CD signals are conveyed to ELL neurons by granule cells, similar to the granule cells of the cerebellum (*Bell et al., 2008*). Anti-Hebbian plasticity at synapses between granule cells and ELL neurons generates negative images that serve to cancel the effects of the EOD on ELL output (*Bell, 1981*; *Bell et al., 1993*; *Bell et al., 1997a*) (*Figure 1A*). However, all past studies

**\*For correspondence:**
ns2635@columbia.edu

**Competing interests:** The authors declare that no competing interests exist.

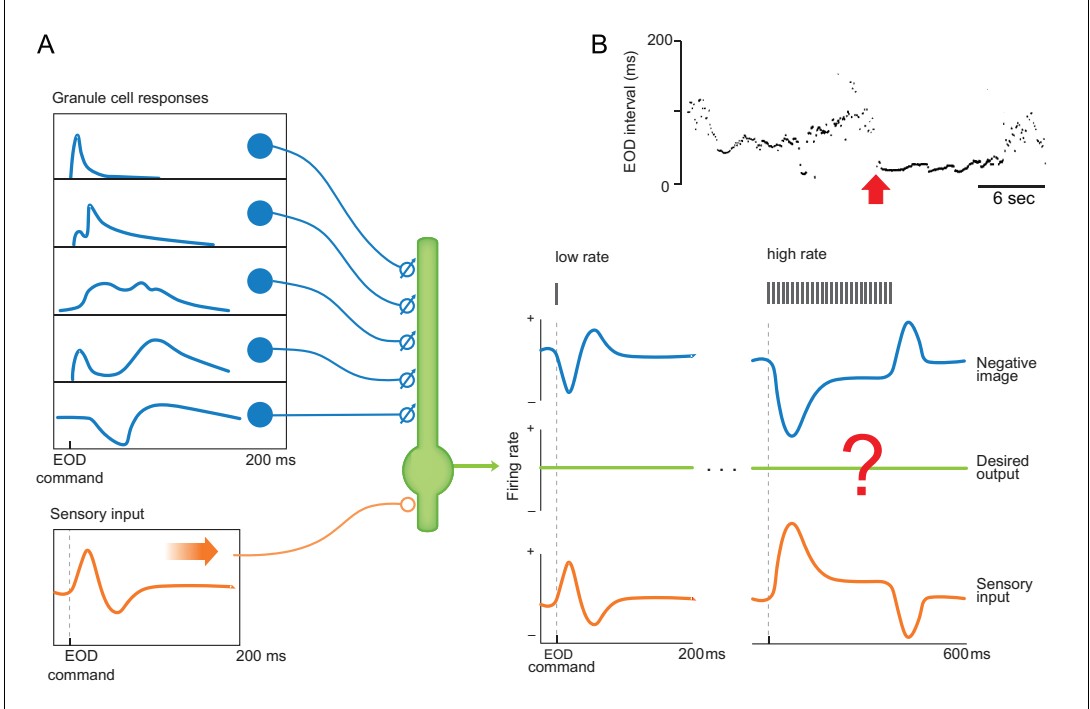

**Figure 1.** Cancelling the effects of the EOD under natural conditions requires generalization. (**A**) Schematic of ELL circuit elements responsible for cancellation of self-generated electrosensory responses. Granule cell corollary discharge responses form a temporal basis (blue trace at left) that is shaped by an anti-Hebbian spike-timing dependent synaptic plasticity rule into a negative image of the predictable sensory response to an EOD (blue trace at right). Signals related to the EOD (orange traces, left and right), along with behaviorally relevant stimuli that the system is meant to detect (not shown), are conveyed by afferent fibers (orange) originating from electroreceptors on the skin. Question mark indicates the process of sensory cancellation being studied. (**B**) A sequence of inter-EOD intervals recorded in a freely swimming mormyrid fish, adapted with permission from *Toerring and Moller (1984)*. Note the wide range of discharge rates and abrupt transition from lower, irregular rates to a high regular rate (arrow). Such transitions highlight the need for negative images to generalize across different EOD rate regimes.

DOI: https://doi.org/10.7554/eLife.44032.002

of negative image formation and sensory cancellation were restricted to periods when fish emitted EOD commands at low, regular rates (~5 Hz). Although this pattern is typical of paralyzed preparations, the fish's actual electromotor behavior is far more dynamic. For example, in freely behaving fish it is common to observe prolonged periods of discharge at low rates (1–5 Hz), while resting or hiding, followed by abrupt transitions to much higher rates (up to 60 Hz) when foraging for prey or exploring a novel object (*Figure 1B*); (*Hofmann et al., 2014*; *Moller et al., 1989*; *Schwarz and von der Emde, 2001*; *Toerring and Moller, 1984*).

During such transitions, negative images learned during low-frequency resting periods should generalize to higher EOD rates. If they do not, passive electrolocation would be degraded at precisely the moment when it would seemingly be most needed. Furthermore, this generalization must be accurate because, at high frequencies, the ringing sensory receptor responses to EODs overlap and, if uncancelled, would continuously interfere with the detection of external stimuli such as prey. Using microstimulation of the EOD motor command pathway to control EOD rate, we show that, indeed, sensory cancellation in ELL output neurons generalizes across EOD rates. In theory, such generalization is expected if electrosensory and corollary discharge responses at high rates were simply the linear sum of the responses at low rates. We show that this is not the case and, instead, identify two key features that, when added to existing models of sensory cancellation in ELL, account for generalization. The first is regularization of synaptic plasticity between granule cells and ELL neurons to prevent overfitting, which is closely related to machine learning approaches to generalization. The second feature, which we support directly by recordings from granule cells and their mossy fiber inputs, involves an approximate matching between the EOD rate-dependence of corollary discharge and electrosensory inputs to ELL neurons.

## Results

### Sensory cancellation in ELL output cells generalizes from low to high EOD rates

We first tested whether sensory cancellation in ELL output cells generalizes across different EOD rates. As in past studies, we used a preparation in which the EOD is blocked by a paralytic, but in which fish are alert and continue to generate the motor commands that normally evoke EODs. The electric field normally generated by the EOD is mimicked experimentally. This preparation permits study of the responses to motor corollary discharge inputs triggered by the EOD command (by turning off the mimic), the sensory response to the artificially produced EOD mimic (by generating the mimic in the absence of an EOD command), and the response to EOD mimics paired with the EOD command. The paired condition replicates the natural situation in which the EOD command evokes an EOD pulse and both electrosensory and corollary discharge pathways are engaged together.

Past studies have shown that the response to locally delivered EOD mimics triggered by the EOD command are cancelled if mimics are paired with commands in this way over ~15 min. For this reason, we will use the term 'learning' to refer to extended periods when EOD mimics are triggered by, and hence paired with, commands. Turning the mimic off after learning reveals that the response to the command alone resembles a negative image of the response to the mimic in the absence of a command (*Bell, 1981*, *Bell, 1982*). As discussed in the Introduction, a limitation of past studies is that cancellation and negative images were only studied at the low EOD command rates (~5 Hz) typical of the paralyzed preparation. We overcame this limitation by using microstimulation of the electromotor command pathway (see Materials and methods) to control the timing and rate of EOD commands (*von der Emde et al., 2000*). Using this approach, we could achieve almost perfect control over the timing of EOD commands at rates up to 50 or 60 Hz.

Extracellular single-unit recordings were made from output cells in the region of the ELL dedicated to passive electrosensory processing—the ventrolateral zone (VLZ). These output neurons are classified into two types, known as E and I cells, according to the polarity of their response to electrosensory stimuli (*Bell, 1981*, *Bell, 1982*). To avoid firing-rate rectification, which complicates quantitative measurements of sensory cancellation, we adjusted the polarity of the EOD mimic to evoke excitatory responses in both E and I cells (see Materials and methods). Consistent with previous findings (*Bell, 1982*; *Enikolopov et al., 2018*), no obvious differences in plasticity were observed between E and I cells and responses were pooled.

To test generalization, we paired evoked commands with EOD mimics at a single constant rate (10 Hz) for a 10–20 min learning period (by which time significant cancellation had occurred; *Figure 2A*, top row) and then probed responses to EOD mimics paired across a range of rates (10, 40, and 60 Hz or 10, 30, and 50 Hz). Responses after learning are reduced across rates even though learning occurred at only the lowest rate, consistent with generalization of cancellation (*Figure 2A*, bottom row, solid lines). An additional set of experiments were performed to provide a benchmark for evaluating the quality of generalization. In this case, the EOD mimic was paired for the same duration but this time learning took place at all the different frequencies that were subsequently tested for cancellation (10, 40, and 60 Hz or 10, 30, and 50 Hz; *Figure 2B*). In this scenario, for which no generalization is required, we expect the system to achieve the best level of cancellation across all rates that can be achieved on the timescale of these experiments. The degree of cancellation, measured as the residual power in the response after learning divided by the power before learning, was comparable in the two sets of experiments (*Figure 2C,D*), indicating excellent generalization.

Past studies have shown that cancellation of predictable electrosensory responses is due to the generation and subtraction of negative images (*Bell, 1981*, *Bell, 1982*). Several observations suggest that the cancellation observed in *Figure 2* is likewise due to the formation of negative images. First, cancellation is unlikely to be due to adaptation of peripheral receptors or neuronal fatigue as we routinely probed responses to the EOD mimic delivered independently of the command both before and after learning (*Figure 2A*, bottom, dashed lines). Reductions in the response to the mimic alone were never observed. Second, in a subset of experiments we probed responses to the command alone across EOD rates after learning only at a low rate. Changes in the response to the command alone resembled a negative image of the response to the mimic sequence (*Figure 2—figure supplement 1*).

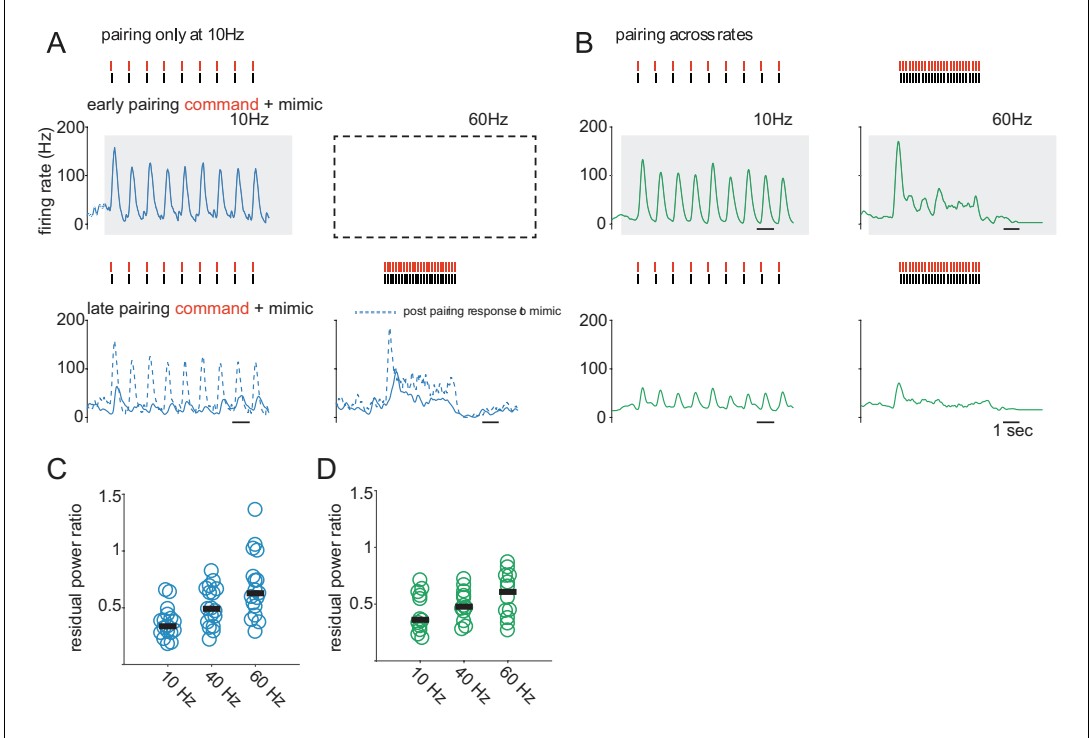

**Figure 2.** Sensory cancellation in ELL output cells generalizes from low to high EOD rates. (**A**) Top, pre-learning response of an ELL output cell to a sequence of mimics triggered by EOD commands at 10 Hz. Shaded box indicates the learning condition. Empty dashed box indicates that no learning was performed at 60 Hz in this series of experiments. Red ticks show the times of EOD commands and black ticks show the times of EOD mimics. Bottom, response of the same cell after learning at 10 Hz. Dashed line is the response of the cell to the EOD mimic presented independent of the command after learning. Note, the response to the mimic is largely cancelled at both 10 and 60 Hz even though learning occurred only at 10 Hz. Responses were also probed at 40 Hz in this cell with similar results (not shown). Scale bar is 1 s. (**B**) Top, pre-learning responses of an ELL output cell to paired EOD command and mimic sequences at 10 and 60 Hz. Shaded boxes indicate that learning took place at both 10 Hz and 60 Hz. Bottom, the response of the same cell after learning. Learning was also conducted at 40 Hz in this cell with similar results (not shown). (**C**) Degree of cancellation at each rate for learning only at 10 Hz, expressed as the ratio of the power of the residual response after learning to the power of the pre-learning response ($n = 17$, median residual power ratios are 0.34, 0.48, 0.63 at 10, 40, and 60 Hz respectively). (**D**) Degree of cancellation at each rate when learning and testing were at the same frequencies of 10, 40, and 60 Hz, expressed as in C ($n = 12$, median residual power ratios are 0.36, 0.49, 0.61 at 10, 40, and 60 Hz respectively).

DOI: https://doi.org/10.7554/eLife.44032.003

The following figure supplement is available for figure 2:

**Figure supplement 1.** ELL neurons form negative images that generalize across EOD rates.
DOI: https://doi.org/10.7554/eLife.44032.004

## Regularized synaptic plasticity partially explains generalization

To gain insights into the mechanisms that support generalization, we adapted a previously developed model of negative image formation and sensory cancellation in the ELL (*Kennedy et al., 2014*). The model ELL neuron receives two classes of inputs. The first is a non-plastic electrosensory input that we simulated by using the recorded response of an ELL output cell to an EOD mimic sequence. This corresponds anatomically to the input onto the basilar dendrites of ELL neurons from interneurons in the deep layers of ELL receiving somatotopic input from ampullary electroreceptor afferents (*Meek et al., 1999*). The second class of inputs consists of a set of ~20,000 model granule cell responses conveying corollary discharge signals related to the EOD command. This corresponds anatomically to excitatory granule cell-parallel fiber synapses onto the apical dendrites of ELL neurons. The model is simplified in that it does not differentiate between two distinct classes of ELL neurons: output cells and medium ganglion (MG) cells (see Discussion). Granule cells are modeled as integrate-and-fire units receiving inputs generated from recorded responses of mossy fibers and unipolar brush cells (the main excitatory inputs to granule cells) to isolated EOD commands (>200 ms

intervals between commands (*Kennedy et al., 2014*). This granule cell model is one component of the full model; the other is a mathematical description of the plasticity of synapses from granule cells to ELL neurons (*Bell et al., 1997a*; *Han et al., 2000*). The anti-Hebbian spike timing-dependent plasticity rule used in the model includes a regularization mechanism to prevent excessively large synaptic weights. Regularization consists of having the synaptic weights decay exponentially toward a baseline value with a time constant of 1000 s, in addition to their modification due to anti-Hebbian plasticity. We refer to this version of the plasticity rule as minimally regularized (see Materials and methods).

To explore mechanisms of generalization using this model, we first needed to extend its granule cell component to simulate high EOD command rates. To begin, we made simple assumptions about how the previously recorded mossy fibers and unipolar brush cells would respond at higher command rates (see Materials and methods). For example, the most common class of mossy fiber inputs, known as early, fire a precisely-timed burst of spikes (duration ~12 ms) at a short delay after each EOD command. To create early mossy fibers responses to command sequences at different EOD rates, we simply repeated the same burst pattern and timing for each command in the sequence (see Materials and methods for assumptions used for other response types; *Figure 3—figure supplement 1*). Later, we will replace these simple assumptions with results derived from experimental measurements of the true EOD-rate dependence of mossy fiber and other inputs. We refer to the granule cell model without these later modifications as the original model.

Using the original model with minimal regularization, we first simulated the generalization experiment in which the system is repeatedly exposed to 10 Hz sequences of EODs for learning and cancellation and then tested at various rates. In agreement with the experimental results, plasticity in the model gradually reduces ELL neuron responses to the EOD, and cancellation is accurate when it is subsequently tested at 10 Hz (*Figure 3A*, lower left). However, in contrast to the experimental results, the model exhibits a dramatic over-cancellation when tested at higher EOD rates (*Figure 3A*, lower right). To determine whether this resulted from a failure of learning or generalization, we simulated the experiments in which the system was trained at all the rates at which it is tested. Under these conditions, the model ELL neuron learns to cancel sensory responses at all the rates tested (*Figure 3B*, lower panels). This indicates that the model can learn to cancel at different EOD rates but fails to generalize low-frequency learning to high EOD rates.

Cancellation performance is comparable between model and data when generalization is not required because training is at both 10 Hz and 60 Hz (*Figure 3C*, data and minimal regularization). Interestingly, when learning is only at 10 Hz, cancelation at 10 Hz is actually better in the minimally regularized model than in the data (*Figure 3D*, data and minimal regularization). This is consistent with overfitting, a feature that is expected to limit generalization. Indeed, when generalization is required, real neurons outperform the minimally regularized model by a large margin (*Figure 3E*, data and minimal regularization). These results show that: (1) our current understanding of ELL circuitry cannot explain the ability of the system to cancel the sensory consequences of EOD sequences in a manner that generalizes from low to high rates and (2) this is not due to an inability of the model system to cancel across rates but is specifically a failure of generalization.

One strategy for improving generalization that is commonly used in machine learning is regularization (*Bishop, 2006*). To enhance regularization, we decreased the decay time constant for the synaptic weights from 1000 s to 10 s. We also changed the value toward which the weights decay from zero to a non-zero baseline (see Materials and methods). The utility of this latter change will be discussed in a later section. We refer to this modified plasticity rule as fully regularized.

When training is performed at both 10 Hz and 60 Hz, cancellation in the fully regularized model is similar to the data and to the minimally regularized model (*Figure 3C*). When trained only at 10 Hz, the fully regularized model matches the data better than the minimally regularized model, presumably by avoiding overfitting (*Figure 3D*). Consistent with this, the fully regularized model exhibits substantially improved generalization across rates compared to the minimally regularized model (*Figure 3E*, minimal and full regularization). However, the fully regularized model still fails to match the generalization performance seen in the data (*Figure 3E*, data and full regularization). These results suggest that the original model is subject to overfitting and that regularization provides a partial solution. However, additional mechanisms are required to match the data. We reasoned that this failure likely reflects the inadequacy of the assumptions we made about how granule cells respond to high-rate EOD commands.

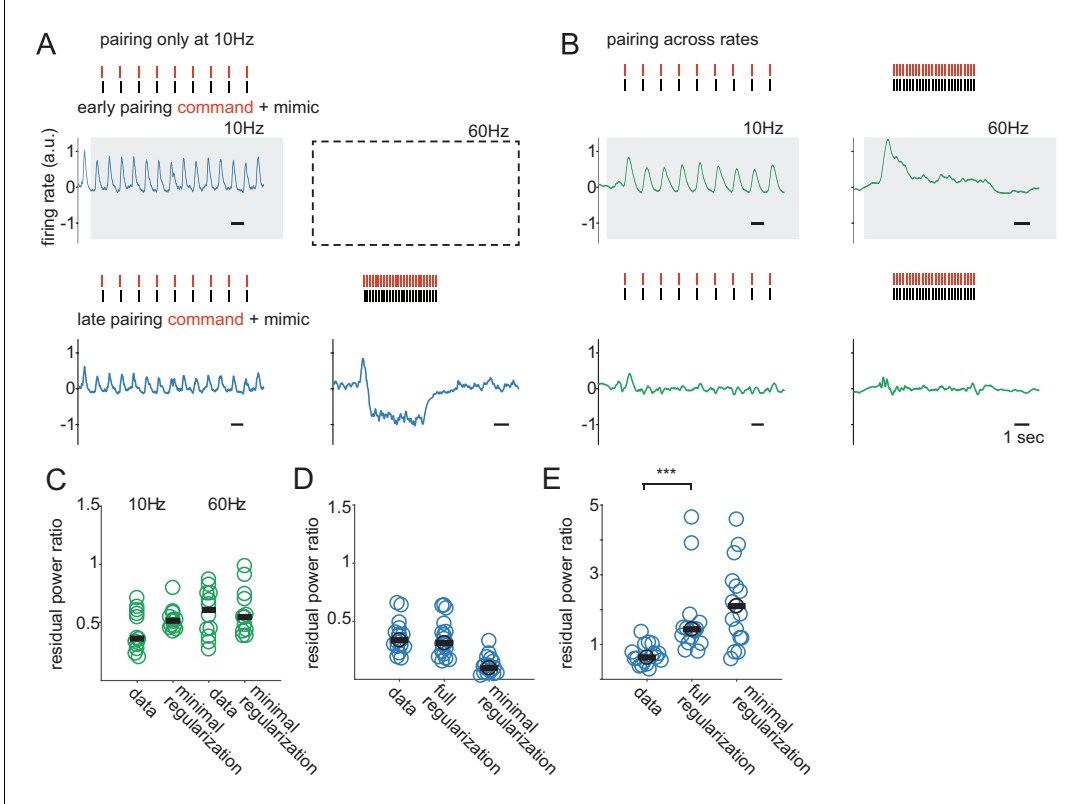

**Figure 3.** Regularization of synaptic plasticity improves but does not fully account for generalization. (**A**) Top, pre-learning response of a model ELL neuron to paired EOD command and mimic sequences delivered at 10 Hz. Shaded box indicates the learning condition. Red ticks show the times of EOD commands and black ticks show the times of EOD mimics. Bottom, response of the model cell after learning. The response to the mimic is largely cancelled at 10 Hz but is dramatically over-cancelled at 60 Hz. (**B**) Top, pre-learning response of a model ELL neuron to paired EOD command and mimic sequences at 10 and 60 Hz. Shaded boxes indicate the learning conditions. Bottom, response after learning. The response is largely cancelled at both 10 and 60 Hz. (**C**) Degree of cancellation at 10 Hz and 60 Hz for model and real cells across rates when training occurred at both rates. (**D**) Degree of cancellation at 10 Hz for real cells and model cells, with full and minimal regularization, when learning was only at 10 Hz. (**E**) Degree of cancellation at 60 Hz for real and model cells, with full and minimal regularization, when learning was only at 10 Hz.

DOI: https://doi.org/10.7554/eLife.44032.005

The following figure supplement is available for figure 3:

**Figure supplement 1.** Modeling the responses of late mossy fibers across EOD command rates.

DOI: https://doi.org/10.7554/eLife.44032.006

## Rate dependence of granule cell corollary discharge responses in vivo

The corollary discharge responses of granule cells provide the 'raw material' from which negative images are sculpted via synaptic plasticity, and hence they are critical for sensory cancellation. Although the granule cell corollary discharge responses used in our model are based on an extensive set of recordings, all of the data was collected in the context of isolated EOD commands (*Kennedy et al., 2014*). As mentioned above, we modeled cancellation at high rates based on assumptions about how mossy fiber inputs to granule cells respond at high command rates. The failure of our original model to match the generalization performance seen in the data, even with full regularization, may indicate that these assumptions are incorrect. To test this, we used whole-cell recordings to characterize corollary discharge responses across EOD rates for 28 granule cells (see Materials and methods).

Most recorded granule cells (21 of 28) exhibited a prominent (~8 mV), short-latency (~2.5 ms) depolarization in response to spontaneously emitted EOD commands. Previous studies have shown that this response type, known as early, is due to mossy fiber input originating from a specific midbrain nucleus that relays electric corollary discharge information (*Bell et al., 1983*). Command-locked hyperpolarizations, indicative of inhibition, were rarely observed, also consistent with past studies.

After characterizing responses to spontaneous commands, microstimulation of the electromotor command pathway was used to evoke trains of 25 commands at rates of 10–60 Hz (for clarity, only responses to low and high rates are shown in the figures). As can be seen in the example traces in *Figure 4A*, command-evoked depolarizations show little or no temporal summation at high command rates, with some cells even exhibiting a relatively hyperpolarized membrane potential at high versus low rates (not shown). Additional examples are shown in *Figure 4—figure supplement 1*. The responses of recorded granule cells contrast with those of the original model, which show pronounced summation and membrane potential depolarization at high rates (*Figure 4—figure supplement 2*, *Figure 5B*).

To quantify the differences between recorded versus model granule cells, we computed the average percentage increase in membrane voltage from 10 Hz to 60 Hz (*Figure 4C*) and the average slope of the line best fit to the membrane voltage of each cell across a 60 Hz train of EOD commands (*Figure 4D*). For the model cells, we generated a distribution by drawing 1000 sets of 28 cells from our model population (matching the 28 recorded cells) and used this to compute both a distribution and a p-value. In each histogram the vertical dashed line shows the value calculated for the set of recorded granule cells. Whereas recorded granule cells showed very little change in their average membrane potential at high command rates, model cells increased their membrane potential substantially (*Figure 4C*). Recorded granule cells have, on average, a negative slope in their membrane potential across 60 Hz trains, whereas model cells have positive sloping membrane potentials (*Figure 4D*), consistent with greater summation in the model versus the recorded granule cells. Clearly the original granule cell model provides a poor description of actual granule cell responses at high EOD command rates.

The shortcomings of the granule cell model we have been using could arise from a mismatch of the model to the biophysical properties of real granule cells, or it could be the result of poorly describing their mossy-fiber and unipolar-brush-cell inputs. To differentiate between these possibilities, we modeled granule cell responses using the same integrate-and-fire description we have been using, but we replaced the computed input to the model cells with experimentally measured inputs. For each granule cell, we fit integrate-and-fire model parameters and, at the same time, inferred its excitatory inputs from the recorded membrane potential. This process was relatively straightforward given that granule cells exhibit large EPSPs, low noise, and receive just a few inputs (*Kennedy et al., 2014*; *Requarth et al., 2014*; *Sawtell, 2010*) (see Materials and methods). We found that the original integrate-and-fire model did a good job of fitting the data provided that we used inputs inferred from data, not the inputs computed in the original model (*Figure 4E* shows the data and model fit for an example cell, also see *Figure 4—figure supplement 1*). We tried a number of augmented models, including features such as synaptic depression, inhibition and conductance-based soma and synapses, but these did not substantially improve the fit compared to the basic current-based integrate-and-fire model with purely excitatory input. This analysis suggested that the failure of the original model (*Figure 3*) to generalize may indeed lie in its failure to accurately represent the EOD-rate dependence of mossy fiber and unipolar brush cell inputs.

To test this more directly, we recorded from mossy fiber axons, unipolar brush cells, and Golgi cells. Criteria for distinguishing between these different elements in neural recordings have been established previously (*Bell et al., 1992*; *Kennedy et al., 2014*; *Sawtell, 2010*). Prominent effects of EOD command rate were observed in early mossy fibers, the most common class of mossy fiber input to granule cells. Consistent with past observations, early mossy fibers fire extremely precise bursts of spikes following after EOD command. After the successive commands in a 10 Hz train they fire exactly the same burst of spikes (*Figure 4F*, top), but on the second command of a 40 Hz or 60 Hz train they drop one or more spikes (*Figure 4F*, middle and bottom). The result of this dropping out is that the average number of mossy fiber spikes fired per command decreases with increasing command rate (*Figure 4G*).

An additional effect observed at high command rates was a decrease in the rate of tonic input to granule cells. Tonic inputs (EPSPs not time-locked to the EOD command) were observed in 19 of 28 granule cells (*Figure 4H*). Similar command-rate dependent decreases in tonic firing were observed in recordings from putative mossy fibers and unipolar brush cells (*Figure 4—figure supplement 3*). Command rate-dependent responses were also found in another previously characterized functional class of granule cell input, known as pause inputs. Pause inputs exhibit a sharply-timed cessation of tonic firing following each EOD command and are believed to correspond to unipolar brush cells

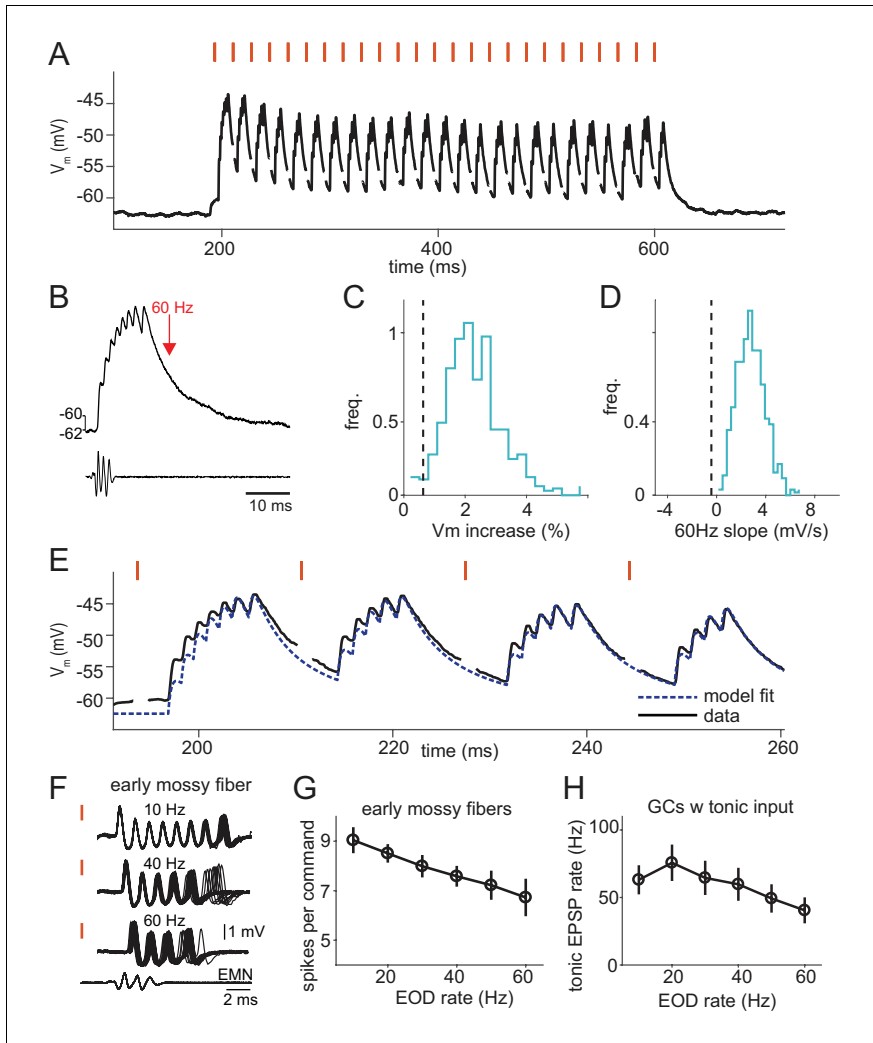

**Figure 4.** Command rate-dependence of granule cells and their mossy fiber inputs. (A) Membrane potential of a granule cell in response to a 60 Hz sequence of 25 EOD commands, with stimulus artifacts removed. Red ticks show the times of EOD commands. (B) The response to a single command at 10 Hz along with the time at which a subsequent command would occur at a rate of 60 Hz (red arrow). Bottom trace is the electromotoneuron volley recorded near the electric organ. Same cell as in (A). (C) Distribution of median percentage increase in maximum membrane voltage from 10 Hz to 60 Hz command rates across $n = 28$ model granule cells. Dashed line shows the experimental value of 0.006 (p=0.03). (D) Distribution of median slope of membrane voltage in response to a 60 Hz sequence for model granule cells, dashed line shows value from the data, $-0.43$ mV/s (p<0.002). (E) Initial portion of the response of the granule cell shown in panel (A). Black lines are data with stimulus artifacts removed, blue dashed line shows a fit using a model granule cell with input spike times inferred from the recorded membrane voltage. (F) Example traces from an early mossy fiber recorded extracellularly in the granular layer. Responses to 25 commands in a 10 Hz (top), 40 Hz (middle), or 60 Hz (bottom) sequence are overlaid. Note the 'dropping' of spikes in the burst at high rates. Bottom trace is the electromotoneuron volley recorded near the electric organ. (G) Average number of spikes fired per EOD command by early mossy fibers (gray, $n = 9$). Symbols show the mean ±S.D. (H) Average firing rate across all inferred tonic mossy fiber inputs to granule cells across EOD command frequencies (mean ±SEM, $n = 19$).

DOI: https://doi.org/10.7554/eLife.44032.007

The following figure supplements are available for figure 4:

**Figure supplement 1.** Example recorded granule cells receiving early input.
DOI: https://doi.org/10.7554/eLife.44032.008

**Figure supplement 2.** Granule cells in the original model fire nonlinearly and this nonlinearity is greater for cells with slower inputs.
DOI: https://doi.org/10.7554/eLife.44032.009

*Figure 4 continued on next page*

*Figure 4 continued*

**Figure supplement 3.** Tonic mossy fibers decrease their firing rate at high EOD command rates.
DOI: https://doi.org/10.7554/eLife.44032.010
**Figure supplement 4.** Pause mossy fibers cease firing at high command rates.
DOI: https://doi.org/10.7554/eLife.44032.011
**Figure supplement 5.** Golgi cells increase their firing rate with increasing EOD command rate .
DOI: https://doi.org/10.7554/eLife.44032.012

(*Kennedy et al., 2014*). Recordings from granule cells and putative UBCs indicate that such pause inputs decrease their firing significantly at higher command rates, often ceasing to fire completely (*Figure 4—figure supplement 4*). Finally, we found that Golgi cells, inhibitory interneurons that synapse onto granule cells, markedly increase their firing with increasing EOD command rate (*Figure 4—figure supplement 5*). Hence, command rate-dependent Golgi inhibition could also contribute to reducing the effect of temporal summation of excitatory inputs in granule cells.

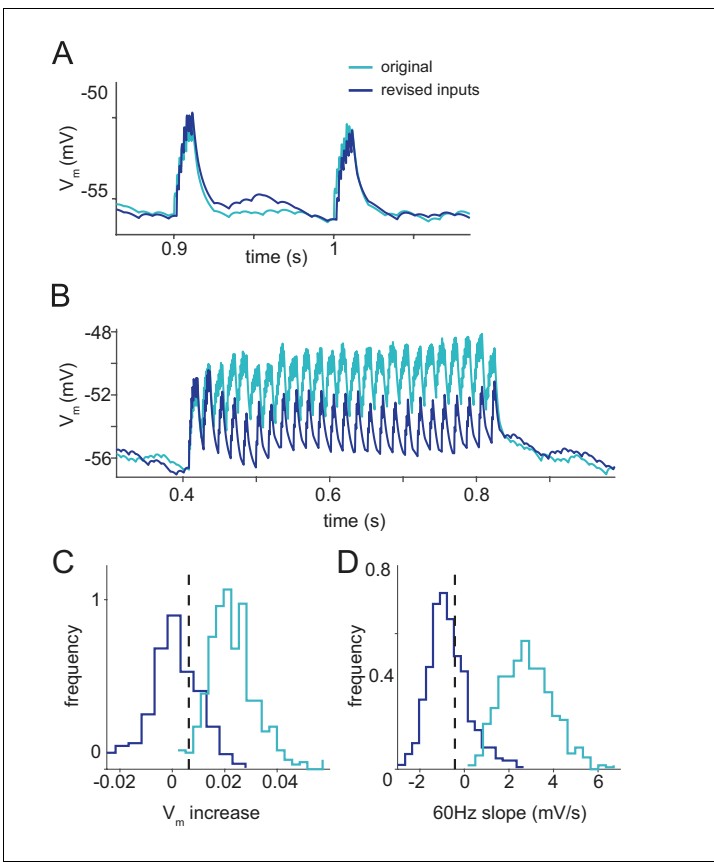

**Figure 5.** Model granule cells with rate-dependent command inputs match recorded granule cells. (**A–D**) Dark versus light blue indicates model cell response with and without rate-dependent inputs matched to the data. (**A**) Response of a model granule cell to two EODs in a 10 Hz command sequence. Note that the cell responds very similarly at this rate with either set of inputs. (**B**) Response of the same model cell as in (**A**), but for a sequence of EOD commands at 60 Hz. Note the qualitatively distinct responses with and without input rate-dependencies at this higher rate. (**C–D**) Distributions of two response statistics for new and old models, dashed lines show the value found in real granule cells. (**C**) Median percentage increase in membrane voltage from 10 to 60 Hz (for old model p=0.03, for new model p=0.72). (**D**) Median membrane potential slope across a 60 Hz train of EOD commands (for old model p<0.002, for new model p=0.81).
DOI: https://doi.org/10.7554/eLife.44032.013

## Model granule cells with rate-dependent command inputs match recorded granule cells

As described above, excitatory inputs to granule cells exhibit EOD command-rate dependencies that are more complex than those assumed in our original model. To determine whether such effects could help to explain generalization in ELL output neurons, we incorporated features of the recorded mossy fibers into a revised model. Specifically, we introduced the rate-dependent dropping out of early mossy fiber spikes and the reduction in tonic mossy fiber firing into the model. The measured rate-dependence of pause mossy fibers was similar to what was assumed in the original model, so no modification was necessary for them. Golgi cells were not considered further because we felt that too little is currently known about the details of Golgi inhibition onto granule cells to incorporate them into the model.

We characterized the effects of these changes by simulating populations of granule cells with and without command rate-dependent inputs. At low command rates model granule cells from the two populations show similar responses (*Figure 5A*). However, at high command rates the two populations differ. Granule cells in the revised model no longer exhibit the increased depolarization at high versus low rates that was observed in the original model granule cells (*Figure 5B*). Examining the statistics we used previously to characterize EOD-rate dependencies in granule cell responses reveals that the inclusion of realistic assumptions regarding mossy fiber inputs dramatically changes the overall character of the granule cell responses in the revised model. The result is model granule cell responses that are clearly more consistent with the subthreshold responses recorded in actual granule cells across EOD rates (*Figure 5C,D*).

## The revised model with full regularization matches the generalization performance of ELL output neurons

Finally, we sought to determine whether the revised granule cell model, combined with fully regularized synaptic plasticity, can explain generalization in ELL output cells. We again simulated the generalization experiment where the system learns with 10 Hz sequences of EODs and cancellation performance is subsequently probed at different EOD rates (*Figure 6A*). The revised model with full regularization shows cancellation across rates that generalizes at a level comparable to the recorded ELL neurons (*Figure 6A,C,D*). To understand the roles of both regularization and EOD rate dependencies, we compared results obtained using the revised model granule cell population (with rate-dependent input) but with the minimally regularized synaptic plasticity rule. In this case, model ELL neurons trained only at 10 Hz exhibited over-cancellation at high EOD rates (*Figure 6B*). Hence the more realistic mossy fiber-granule cell model, on its own, is also insufficient to explain generalization (*Figure 6B,C,D*).

Further examination of the model suggests a hypothesis regarding how regularized synaptic plasticity and rate-dependent mossy fiber inputs work together to support generalization. The form of regularized synaptic plasticity we have used involves a decay of each synaptic weight toward a constant non-zero value. Increasing the strength of this regularization decreases the variance of the learned weights because synaptic weights from different granule cells are constrained to be similar to this value and hence to one another (*Figure 6—figure supplement 1*). This means that, with strong regularization, the learned negative image is constrained to be approximately proportional to the mean response of the granule cell population. This average shape is, in turn, affected strongly by the rate-dependence of inputs to granule cells. As we have shown, in the absence of realistic mossy fiber rate-dependencies the mean model granule cell response has an increasing profile across a 60 Hz train, whereas, with the actual mossy fiber rate dependence, the mean granule cell response has a decreasing profile.

Notably, the sensory responses of ELL neurons to high rate trains of EOD mimics (prior to cancellation) also exhibit a decreasing profile (*Figure 2A*, *Figure 6—figure supplement 2*). To determine the origin of such responses, we performed a separate set of extracellular recordings from ampullary electroreceptor afferents (the source of electrosensory input to ELL neurons). Ampullary afferent firing rate also exhibited a decreasing profile at high EOD rates (*Figure 6E*, *Figure 6—figure supplement 3*). Responses of ampullary afferents to isolated EOD pulses consist of a firing rate increase followed by a reduction below baseline and in some cases additional smaller waves of increased and decreased firing resembling a damped oscillation (*Figure 6—figure supplement 3D*)(*Bell and*

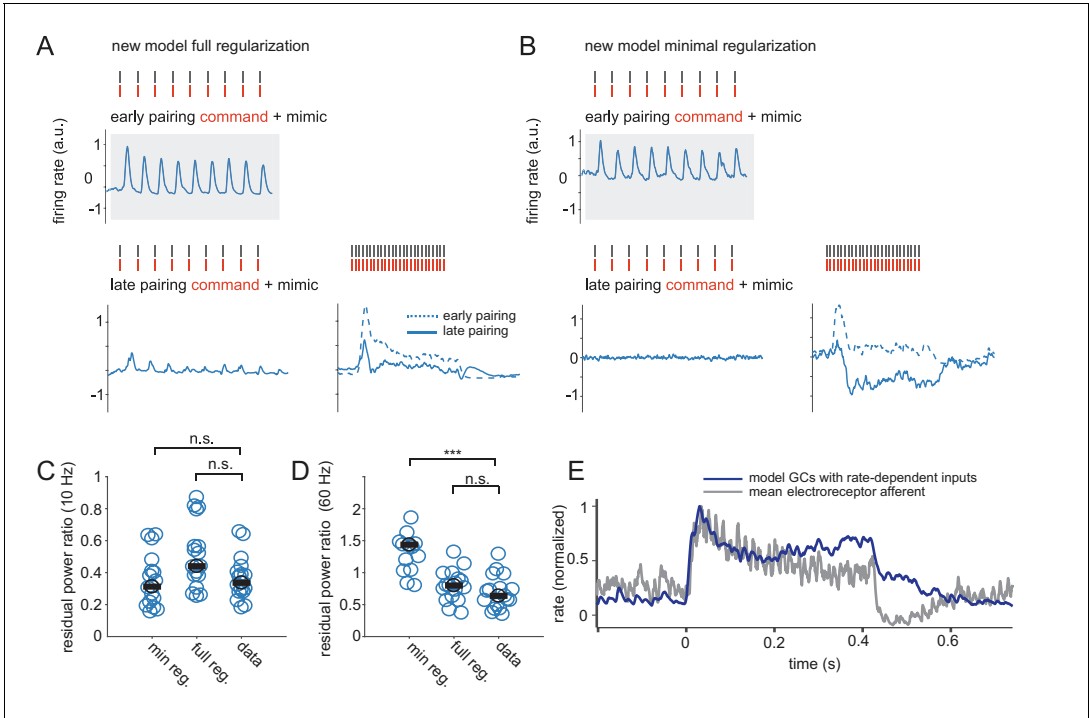

**Figure 6.** A revised ELL model accounts for generalization in ELL neurons. In all panels dashed blue traces are the sensory response to be cancelled and solid blue traces are the response to the paired EOD command plus mimic sequences. Learning occurs only at 10 Hz as indicated by grey boxes. Red ticks show the times of EOD commands and black ticks show the times of EOD mimics. (A) Top, pre-learning response of a revised model output cell with full regularization to paired EOD command and mimic sequences delivered at 10 Hz. Bottom, response of the same cell after learning. Note, the response to the mimic is largely cancelled at both 10 and 60 Hz. (B) Top, pre-learning response of a revised model output cell with minimal regularization to paired EOD command and mimic sequences delivered at 10 Hz. Bottom, response of the same cell after learning. Note, the response to the mimic is largely cancelled at 10 Hz but is now over-cancelled at 60 Hz. (C) Level of cancellation achieved at 10 Hz across different model and real ELL cells is similar (p=0.72 for minimally regularized model versus data; p=0.62 for fully-regularized model versus data, Wilcoxon signed rank test). (D) Similar to C but showing the level of cancellation achieved at 60 Hz, (p<0.001 for minimally regularized model versus data; p=0.38 for fully-regularized model versus data, Wilcoxon signed rank test). (E) Dark blue, mean spiking response of model granule cells with input rate dependencies; grey, mean response of electroreceptor afferents, both at 60 Hz EOD rate. Note the similarity in shape.

DOI: https://doi.org/10.7554/eLife.44032.014

The following figure supplements are available for figure 6:

**Figure supplement 1.** Stronger regularization of synaptic plasticity restricts negative images to be proportional to the mean granule cell response and decreases variance in synaptic weights.

DOI: https://doi.org/10.7554/eLife.44032.015

**Figure supplement 2.** Rate-dependence of ELL output cell responses to the EOD.

DOI: https://doi.org/10.7554/eLife.44032.016

**Figure supplement 3.** Rate-dependence of ampullary afferent responses to the EOD.

DOI: https://doi.org/10.7554/eLife.44032.017

*Russell, 1978*). Estimating the impulse response of an ampullary afferent from its average response to a single EOD mimic and then convolving this impulse response with a sequence of EOD mimics yielded a reasonable approximation to the observed responses (*Figure 6—figure supplement 3A*, red lines). Hence the decaying profile of the sensory response to high-rate sequences of EODs as well as the inhibitory rebound at the end of such sequences are expected features of a linear system with an impulse response resembling a damped oscillation.

In summary, these results suggest that generalization across EOD discharge rates may be achieved in the ELL by combining two features: (1) a form of plasticity that encourages low variance in learned weights, forcing the negative image to be close to the mean granule cell response and (2) mossy fiber rate dependencies that ensure that the mean granule cell corollary discharge response has a shape that approximates the sensory signal to be cancelled. Together, these two features may

allow accurate negative images to be generated across a wide range of EOD rates for which no previous learning has taken place.

## Discussion

### Functional significance of generalization in the ELL

Past work on negative image formation and sensory cancellation in mormyrid fish has been restricted to one particular behavioral regime, namely, periods when EOD rates are low and regular. However, the rate and timing of EODs are under voluntary control and vary widely during both electrocommunication and active electrolocation (*Hofmann et al., 2014*; *Moller et al., 1989*; *Schwarz and von der Emde, 2001*; *Toerring and Moller, 1984*). This suggests that negative images learned over periods of minutes or hours at low EOD rates (e.g. while the fish is inactive) must generalize when the fish transitions to a high EOD rate (e.g. during foraging, fleeing, exploring a novel object, or interacting with a conspecific). If generalization did not occur in such instances, the passive electrosensory system would be vulnerable to self-generated interference during the periods when it would be needed the most. Behavioral studies suggest that multiple senses (including both the passive and active electrosensory systems) are used in concert to detect prey (*von der Emde and Bleckmann, 1998*). In light of these considerations, our observation that the cancellation performance of ELL neurons generalizes accurately supports and substantially extends the ethological relevance of negative images for passive electrolocation in mormyrid fish. A caveat is that our study focused on generalization in only one specific set of circumstances, that is an abrupt transition from low to high rates. Although this allowed us to focus on in-depth analysis of the mechanisms of generalization, important questions remain about how such mechanisms operate under more natural circumstances, for example in the context of the more complex and variable EOD interval patterns characteristic of behaving fish.

Our results are likely relevant to a number of other behavioral contexts and brain structures in which the cancellation of self-generated sensory inputs is known to occur. Negative images have been described in the active electrosensory system of mormyrid as well as gymnotid fish where they serve to cancel the effects of movements of the fish's body as well as spatially redundant electrosensory signals resulting from interactions with conspecifics (*Bastian, 1996*; *Requarth et al., 2014*; *Requarth and Sawtell, 2014*). Movements of the tail, for example, generate reafference by changing the position of the electric organ (located in the tail) relative to electroreceptors on the head and body. In the passive electrosensory system of elasmobranchs (the group that includes sharks and skates), negative images cancel the effects of swimming movements and respiration (*Bodznick et al., 1999*). Cancellation of self-generated inputs has also been described in related cerebellum-like structures associated with the mechanosensory lateral line system in fish and the auditory system in mice (*Montgomery and Bodznick, 1994*; *Singla et al., 2017*). In all of these cases, generalization is expected to be vital in assuring that negative images remain accurate across different behavioral and/or environmental contexts.

### Mechanisms of generalization

Using a combined experimental and theoretical approach we identified two features that, when added to existing models of ELL, were sufficient to explain how negative images learned at one rate generalize to another. The first element was that synaptic plasticity from granule cells to ELL neurons be appropriately regularized. Regularization of learned parameters is ubiquitous in machine learning as a technique to prevent overfitting, or the learning of parameters that fit the idiosyncrasies and noise present in training data and therefore do not generalize well to new data (*Bishop, 2006*). Consistent with this, in an ELL model lacking adequate regularization we found that ELL neurons could learn negative images at low EOD rates, however, cancellation at high (untrained) rates was poor. Although we do not have direct evidence for such regularization of synaptic plasticity in ELL, we note that there are a number of candidate mechanisms described in other systems. For example bounded synaptic strengths (*Amit and Fusi, 1992*), discrete synaptic weights (*O'Connor et al., 2005*; *Petersen et al., 1998*), synaptic scaling (*Turrigiano, 2008*), coupling of synaptic changes between nearby synapses (*Engert and Bonhoeffer, 1997*), synaptic competition (*Miller, 1996*), and various sources of noise (*Basalyga and Salinas, 2006*), could all act as forms of regularization even if

they are simply due to constraints on the system or have additional purposes. In our model we found that a constant decay of the strength of each synapse towards a baseline value worked best. This rule has the appealing property of being implementable locally at each synapse. However, our rule does require an explicit setting for the regularization decay rate. This parameter could itself be learned over a longer timescale, which would be a form of meta-plasticity (*Abraham and Bear, 1996*) or meta-learning (*Doya, 2002*). To our knowledge, little is currently known in any biological system regarding whether and how synaptic plasticity is regularized or about whether such regularization plays a role in generalization. Addressing these questions is an interesting challenge for future research that may be aided by emerging methods for directly visualizing morphology, activity, and synaptic proteins at the level of dendrites and spines (*Roth et al., 2017*).

The second feature we identified as important for generalization is an approximate matching between the EOD rate dependence of electrosensory inputs to ELL output neurons and the rate dependence of the summed corollary discharge input that an output cell receives via the granule cells. In vivo recordings from ampullary electroreceptor afferents, ELL output neurons, mossy fibers, and granule cells provided direct evidence for such matching. The temporal dynamics of granule cell corollary discharge responses across EOD rates are, on average, much more similar to those of electroreceptor afferents than expected based on past recordings and modeling of granule cell responses to isolated EOD commands. This matching appears to be achieved via a variety of previously unknown EOD command rate dependencies in the inputs to granule cells. So-called early mossy fibers are known from previous studies to fire a highly-stereotyped burst of action potentials following each EOD command (*Bell et al., 1992*). We found that the number of spikes in such bursts declines progressively with increases in the command rate. The multiple spikes in the burst seem redundant in the context of isolated EODs. Why would multiple spikes be needed to signal the time of occurrence of an EOD command? The present work suggests that rate-dependent grading of such bursts conveys information that is important for generalization.

We mainly focused on the command rate-dependence of so-called early mossy fiber inputs because these inputs are by far the most frequently encountered in our blind recordings. However, the command-rate dependence of less common elements such as unipolar brush cells and Golgi cells was also qualitatively consistent with the proposed matching. Determining the relative importance for generalization of these different sources of command-rate dependence (i.e. mossy fibers, unipolar brush cells, and Golgi cells) is difficult given that we lack methods for selectively targeting them for recordings or manipulations. We also cannot rule out the importance for generalization of other circuit elements not studied here and for which we lack sufficient physiological data under conditions of different EOD rates. Our model (like all past models of the mormyrid ELL) does not distinguish between two distinct classes of ELL neurons: glutamatergic output cells versus GABAergic MG cells which inhibit output cells. MG cells occupy an analogous position in the circuitry of the mormyrid ELL as Purkinje cells in the teleost cerebellum and cartwheel cells in the dorsal cochlear nucleus (*Bell, 2002*; *Bell et al., 2008*). Importantly, both MG and output cells integrate electrosensory and corollary discharge input and both exhibit anti-Hebbian plasticity (*Bell et al., 1997a*; *Bell et al., 1993*; *Meek et al., 1996*; *Mohr et al., 2003*). However, it is presently unknown, even in the context of low EOD rates, how MG cells contribute to sensory cancellation and negative image formation. Our model also omits molecular layer interneurons and does not distinguish between E- and I-type output cells. Constructing a more complete and realistic model that includes these additional features is a focus of ongoing experimental and theoretical studies of the mormyrid ELL.

Generalization of negative images could be accomplished quite simply if both electrosensory and corollary discharge signals had a linear dependence on EOD rate. Responses of ampullary electroreceptor afferents, indeed, appear to exhibit a roughly linear dependence on EOD rate (*Figure 6—figure supplement 3*). However, recordings from granule cells in mormyrid fish (*Kennedy et al., 2014*; *Requarth and Sawtell, 2014*; *Sawtell, 2010*), as well as studies of cerebellar granule cells in mammals (*Barmack and Yakhnitsa, 2008*; *Chabrol et al., 2015*; *Chadderton et al., 2004*; *Ruigrok et al., 2011*), suggest that granule cells exhibit markedly nonlinear properties, including prominent rectification and burst firing. In our initial modeling we found that even when electrosensory and mossy fiber inputs both varied approximately linearly with EOD rate and inputs were summed linearly by model granule cells, the model failed to match the generalization performance seen in real ELL neurons. One reason for its failure is the nonlinearity introduced by the firing rate threshold of the granule cells. Whenever a threshold is applied, portions of a signal that are

subthreshold at low repetition rates can become supra-threshold at higher rates due to temporal summation, resulting in nonlinear responses (*Figure 4—figure supplement 2*). Rather than linearizing the granule cell population response, the EOD rate dependencies we found in mossy fiber inputs to granule cells actually introduce additional nonlinearities on top of the threshold linearity. It is the summed effect of these nonlinearities across the granule cell population that guarantees that an approximate negative image is always available in the scaled mean of the population activity. This may be a useful principle employed by other neural systems - encoding of approximate solutions across contexts in a robust manner, in this case through a simple average population activity, allowing flexible learning while maintaining information that supports generalization.

## Connections to generalization in other systems

The issue of generalization has been explored in the gymnotid ELL in the context of cancellation of spatially redundant electrosensory signals, such as those generated by tail movements or conspecifics (*Bol et al., 2011*; *Mejias et al., 2013*). Such cancellation is similar to that in the mormyrid ELL in that it is mediated by anti-Hebbian plasticity at synapses between granule cells and ELL neurons (*Harvey-Girard et al., 2010*). However, cancellation in the gymnotid ELL is driven by proprioception or electrosensory feedback to granule cells rather than by corollary discharge (*Bastian et al., 2004*; *Chacron et al., 2005*). In vivo recordings from ELL neurons in gymnotids demonstrated that cancellation remains accurate over a wide range of stimulus contrasts (as might be produced by conspecifics at different distances) (*Mejias et al., 2013*). Modeling was used to show how learning at one contrast could generalize to higher or lower contrasts, despite numerous nonlinearities in the system. Interestingly, features of the model identified to be important for such generalization are related to those described here for the mormyrid ELL, including granule cell response properties and a slow decay in parallel fiber synaptic strength (*Mejias et al., 2013*; *Lewis and Maler, 2004*), which can be considered a form of regularization. Although responses of granule cells have not yet been measured in vivo in gymnotids, several lines of evidence suggest that they are important in relation to the specificity and generalization of learning in the gymnotid ELL (*Bol et al., 2011*; *Mejias et al., 2013*).

A role for cerebellar granule cells in generalization has been suggested based on studies of motor learning. Adaptation of the vestibulo-ocular reflex (VOR) shows various patterns of generalization and specificity when training and testing are carried out at different head rotation frequencies or static head tilts (*Boyden et al., 2004*). Under some experimental conditions, VOR learning has been shown to be quite specific to the training context (*Baker et al., 1987*; *Yakushin et al., 2000*). Such specificity can be explained by models in which learning is mediated by changes in granule cell inputs conveying highly-specific representations of the training context–for example, granule cells that fire for specific combination of head rotation and head tilt. Such hypotheses have not been directly tested in the context of the VOR or other forms of motor learning, however, numerous lines of evidence support the existence of highly-selective granule cell representations of this sort (*Chabrol et al., 2015*; *Huang et al., 2013*; *Ishikawa et al., 2015*; *Sawtell, 2010*). Generalization of VOR learning is also observed under some circumstances, for example when training at a high head rotation frequency and testing on a lower frequency (*Boyden et al., 2004*). Broader tuning in granule cells could underlie generalization in such cases. Studies of generalization of VOR learning may be informed by recent characterizations of the statistics of vestibular input during natural behavior in primates and rodents (*Carriot et al., 2014*, *Carriot et al., 2017*).

It has been suggested that patterns of generalization in human motor learning, such as adaptation to force fields in reaching, can be explained by the tuning of a set of basis elements (*Donchin et al., 2003*; *Ghahramani et al., 1996*; *Shadmehr and Mussa-Ivaldi, 1994*). Given their large numbers and the well-characterized plasticity of their parallel fiber synapses, granule cells are a natural candidate for such elements. To our knowledge, the present study is the first to directly relate responses of granule cells recorded during a learning task to generalization.

# Materials and methods

## Experimental preparation

All experiments performed in this study adhere to the American Physiological Society's Guiding Principles in the Care and Use of Animals and were approved by the Columbia University Institutional Animal Care and Use Committee, protocol AAAW4462. Mormyrid fish (7-12 cm in length) of the species *Gnathonemus petersii* were used in these experiments. Surgical procedures to expose the brain for recording were identical to those described previously (*Bell, 1982*; *Enikolopov et al., 2018*; *Sawtell, 2010*). Gallamine triethiodide (Flaxedil) was given at the end of the surgery (~20 μg / cm of body length) and the anesthetic (MS:222, 1:25,000) was removed. Aerated water was passed over the fish's gills for respiration. Paralysis blocks the effect of electromotoneurons on the electric organ, preventing the EOD, but the motor command signal that would normally elicit an EOD continues to be emitted spontaneously at rates of 2-5 Hz. The timing of the EOD motor command can be measured precisely allowing the central effects of corollary discharge inputs to be observed in isolation from the electrosensory input that would normally result from the EOD. In a few experiments, recordings from electroreceptor afferents were performed in unparalyzed fish anesthetized with metomidate which leaves the fish's EOD intact (*Engelmann et al., 2006*).

## EOD command stimulation

We controlled the EOD motor command rate by targeting a concentric bipolar stimulating electrode (FHC, Bowdoin, ME) to the axon tract connecting the precommand nucleus to the EOD command nucleus, located near the ventral surface of the brainstem. The electrode was inserted at the midline through the corpus cerebellum just anterior to ELL (angled 22 degrees caudally in the sagittal plane) and lowered into the brain using a hydraulic manipulator until commands could be evoked by a strong stimulus (0.2 ms duration; 50 μA). The depth of the electrode was then fine-tuned until commands could be reliably evoked at short latencies by single pulses using minimal current (typically 5-15 μA). In most cases, such stimulation gave near perfect control over the timing of the EOD command. Occasionally, stimulation failed to evoke a command during high rate trains or the fish discharged spontaneously during a low rate train. However, these errors were easy to detect and were sufficiently infrequent that they were deemed negligible. Finally, microstimulation-evoked corollary discharge responses were indistinguishable from those evoked by the fish's spontaneous commands at the level of field potentials, mossy fibers, and granule cells.

## Electrophysiology

The EOD motor command signal was recorded with an electrode placed over the electric organ. The command signal is the synchronized volley of electromotoneurons that would normally elicit an EOD in the absence of neuromuscular blockade. The command signal lasts about 3 ms and consists of a small negative wave followed by three larger biphasic waves. The latencies of central corollary discharge or command-evoked responses were measured with respect to the negative peak of the first large biphasic wave in the command signal.

Extracellular recordings from the VLZ were made with glass microelectrodes filled with 2M NaCl (8-30 MΩ). Consistent with previous studies, ampullary afferents were encountered in the deeper layers of ELL (medial in our penetrations) and were characterized by highly regular spontaneous firing at ~50 Hz, the absence of any response to the EOD motor command, an excitatory responses to a stomach negative EOD mimic pulse (0.2-2 ms duration), and strong responses to small (<1 μA), long duration (10-100 ms) electrosensory stimuli. Output cells were encountered in more superficial layers (on penetrations slightly lateral to those in which afferents were encountered) and were characterized by much lower and more irregular firing rates than ampullary afferents. E-cells showed increased firing in responses to a stomach negative EOD mimic pulse, while I-cells showed decreased firing. None of the E- or I-cells included in our analysis exhibited two distinct action potential waveforms, the hallmark of the other major cell type of in VLZ, the medium ganglion cells (*Bell et al., 1997a*; *Bell et al., 1993*). Hence these recordings are presumed to be from the efferent neurons of ELL.

Extracellular recordings from mossy fibers in EGp and in the paratrigeminal command-associated nucleus were made using glass microelectrodes filled with 2M NaCl (40-100 MΩ). For in vivo whole-

cell recordings from EGp neurons patch electrodes (9-15 MΩ) were filled with an internal solution containing, in mM: K-gluconate (122); KCl (7); HEPES (10); Na2GTP (0.4); MgATP (4); EGTA (0.5), and 0.5% biocytin (pH 7.2, 280-290 mOsm). No correction was made for liquid junction potentials. Only cells with stable membrane potentials more hyperpolarized than -50 mV and access resistance < 100 MΩ were analyzed. Membrane potentials were filtered at 3-10 kHz and digitized at 20 kHz (CED power1401 hardware and Spike2 software; Cambridge Electronics Design, Cambridge, UK).

## Pairing experiments

Cancellation and negative image formation was tested in E- and I-cells by pairing EOD commands with an EOD mimic pulse (0.2-2 ms wide square pulses; 1-5 µA) delivered at the delay at which the EOD would normally occur (4.5 ms after the EOD command). This delay was fixed and independent of command rate in our experiments. This is assumed to be the case under natural conditions as well, although to our knowledge, this has never been directly shown. EOD mimics were delivered using a dipole electrode positioned < 2 mm from the skin within the unit's receptive field. These methods are the same as those used previously to characterize negative images and sensory cancellation in the context of low command rates. In I cells, the EOD mimic often drove the firing rate to zero, making it difficult to quantify cancellation. To avoid this firing rate rectification, we reversed the EOD mimic polarity when recording from I cells, such that they responded with excitation instead of inhibition. This response reversal is due to known properties of ampullary electroreceptor afferents, which increase (or decrease) firing above (or below) their baseline rate for stimuli that make the pore of the receptor positive (or negative) with respect to the basal face within the body. Past studies have commonly used this approach to demonstrate the specificity of negative image formation in the VLZ by performing multiple pairing in the same neuron using opposite stimulus polarities (*Bell, 1981*, *Bell, 1982*; *Enikolopov et al., 2018*). Negative images are invariably observed for both stimulus polarities in such experiments, that is responses to the corollary discharge alone after pairing are opposite in sign to the response to the stimulus during pairing. Systematic differences between negative images and cancellation in E versus I cells or for mimics of opposite polarities have never been noted, justifying this approach to avoid rectification.

Two types of pairing experiments were conducted. For the first type, pairing was performed across a range of rates (10, 40, and 60 Hz or 10, 30, and 50 Hz). Cancellation was assessed by comparing responses early and late during pairing which lasted 10-20 minutes. Negative images were assessed in a subset of cells by comparing responses to the command alone across rates before versus after pairing. Responses to identical trains of electrosensory stimuli presented independent of the command were also tested for each cell. In some cases, multiple pairings were conducted in the same cell after allowing 10-15 minutes for recovery from the effects of prior pairing. The second type of experiment was the same as described above except that pairing was only conducted at 10 Hz. Cancellation was assessed in these experiments by briefly (60-100 sec) probing responses at all three rates before and immediately after pairing at 10 Hz.

## Linear model of electroreceptor sequence responses

To test whether electroreceptor afferent responses to EOD sequences could be approximated as linear, we estimated the impulse response kernel, $K(t)$, of each recorded unit from its response to an isolated EOD mimic. We first computed the average firing rate evoked by isolated EOD mimics (those separated by at least 150 ms). We treated this as an estimate of the impulse response of the recorded unit. To compute the predicted linear response, $L(t)$, we convolved this kernel with a series of delta functions centered on the times of the EOD mimics:

$$L(t) = \Sigma_i K(t) * \delta(t_i - t)$$

where $t_i$ are the times of the EOD commands in the sequence.

## Quantification of cancellation and generalization

To quantify cancellation and generalization, the degree of cancellation, $C$, was measured as the ratio of the total variance of the response to a sequence starting at time $t_{start}$ and ending at time $t_{end}$, to an EOD command plus mimic sequence post pairing, $r_{post}(t)$, to that pre pairing, $r_{pre}(t)$:

$$C = \frac{\int_{t_{\text{start}}}^{t_{\text{end}}} \left( r_{\text{post}}(t) - \langle r_{\text{post}} \rangle \right)^2 dt}{\int_{t_{\text{start}}}^{t_{\text{end}}} \left( r_{\text{pre}}(t) - \langle r_{\text{pre}} \rangle \right)^2 dt}$$

## Modeling granule cells

Our general approach to modeling granule cells follows that used previously (*Kennedy et al., 2014*). We generate model granule cell populations by random mixing of mossy fiber inputs, as described below. To extend this model to the case of different EOD command rates we also directly fit integrate-and-fire models to recordings of real granule cell responses, inferring the mossy fiber inputs at the same time. We then use information about rate dependencies in these mossy fiber inputs gleaned from this fitting procedure as well as from direct recordings of mossy fibers to show that rate-dependent changes in the inputs to granule cells can account for their responses to EOD command sequences. This information was then used to update the granule cell population model. The following sections describe these different modeling steps in more detail.

## Fitting granule cell voltage responses to EOD command sequences

When fitting models to real granule cell data we first removed stimulus artifacts caused by electromotor command nucleus stimulation as well as any spikes using a simple threshold on the gradient of the membrane voltage. We found that a gradient threshold of 1.8 mV/ms worked well. We used multiple methods and models to fit a set of 28 intracellularly recorded granule cell responses to EOD command sequences from 10 to 60 Hz. The basic model was an integrate-and-fire model with current based synapses. The parameters of the model were the membrane time constant, the leak potential, and synaptic parameters. Each cell could receive two inputs. Each input had the following parameters, a fast and a slow time constant, and a fast and a slow weight. This accounts for the fact that granule cell EPSPs often show a combination of fast and slow components. We allowed two inputs to permit both command-associated inputs and command-independent tonic inputs. The response of a model granule cell was given by

$$\tau_{\text{m}} \frac{dV}{dt} = E_l - V + \Sigma_{i,j} E_i(t) \delta \left( t - t_{ij} \right)$$

where $t_{ij}$ is the time of the $j$-th spike of the $i$-th input and the synaptic kernels for each input are given by

$$E_i(t) = \frac{1}{\tau_i^{\text{fast}}} w_i^{\text{fast}} e^{-\frac{t}{\tau_i^{\text{fast}}}} + \frac{1}{\tau_i^{\text{slow}}} w_i^{\text{slow}} e^{-\frac{t}{\tau_i^{\text{slow}}}}. \tag{1}$$

To fit granule cell models, we further needed to estimate the times of input spikes for each cell. We used multiple methods to make this input inference, all of which gave similar results. The first method was a wavelet-based detection method. We computed the continuous wavelet transform of the membrane voltage at 16 scales from 800 to 2000 Hz, using the MATLAB Wavelet Toolbox. We searched for times where the wavelet transform exhibited peaks at multiple scales and considered these times putative input spike times. We then combined peak locations into single putative input times if they were closer than 0.5 ms together. We then visually validated all of the data by checking if the input times detected by this procedure corresponded to clear upticks in the membrane voltage. We made corrections when it appeared an EPSP had occurred and then used both the corrected and uncorrected input times when fitting models and compared results. The qualitative results described in the main text did not depend on the input details at this level of accuracy. We used two different methods for estimating the parameters of granule cell models. Our results did not depend on which method was used. The first method was to use the putative input times we found, assume these were the actual input times to the granule cells, and then use least squares minimization to find the optimal parameters of the granule cell model, given these input times. The second method was to use these putative input times as an initialization for an MCMC method which then generated joint samples from the posterior distribution of granule cell model parameters and input times. We initialized the input times based on those found by the wavelet method. We then used Gibbs sampling to sample model parameters after a burn-in of 500 sweeps. Approximate sampling of input times was achieved by allowing the following moves: an input spike could be jittered

around its current location, an input spike could be removed, and an input spike could be added. We placed priors on the total number of spikes based on the estimated number detected by the wavelet method to prevent the addition of many extra spikes. We also placed hard bounds on the parameters so that synaptic weights were always positive.

## Basic granule cell model

As in previous work (*Kennedy et al., 2014*), we generated populations of model granule cells from a random mixing procedure based on the following assumptions. Each cell receives input from classes early (E), medium (M), late (L), pause (P) or tonic (T). (i) Each granule cell has three sites for mossy fiber synaptic inputs. (ii) The probabilities of a given input being of E, M, L, P and T type are given by $P_e$, $P_m$, $P_l$, $P_p$ and $P_t$, with $P_e + P_m + P_l + P_p + P_t \leq 1$. (iii) The type of input received at one mossy fiber-granule cell synapse is independent of that received at any other synapse. We used input type probabilities as calculated previously based on fits to individual granule cells (*Kennedy et al., 2014*).

We introduced two sources of variability. We included trial to trial variability in the peak height of recorded single EPSPs from a normal distribution with $\sigma = 0.224$ mV; during simulation of model granule cells, we sampled this distribution for each mossy fiber spike. Some granule cells further receive tonic mossy fiber inputs in addition to corollary discharge inputs. These inputs fire at high rates, independent of the EOD command. We included tonic input as previously, based on 72 tonic mossy fiber recordings.

For each model granule cell we randomly determined whether each potential connection to that model cell received early ($P_e = 0.425$), medium ($P_m = 0.075$), late ($P_l = 0.05$), pause ($P_p = 0.05$), tonic ($P_t = 0.157$) or no input ($P_n = 0.243$), as in previous work. We then chose a particular mossy fiber response of the previously-determined class as the source of that input; we assumed that a connection is equally likely to be from any of the mossy fibers within a given class. These steps constitute the basic procedure for modeling populations of granule cells. The mossy fiber recordings we use to generate granule cells were based on responses to single EOD commands. To model the responses of these cells to EOD command sequences we needed to choose a method for predicting the responses of each mossy fiber to sequences of EOD commands. The following sections describe how this was achieved.

In the model granule cells we used synapses with fast and slow components. We used the same synapse model described above when fitting responses of real granule cells. The synaptic dynamics were described by *Equation 1*. When generating model granule cell populations we had to choose values for the four parameters $\tau_{fast}$, $w_i^{fast}$, $\tau_{slow}$, and $w_i^{slow}$. These were chosen by fitting Gamma distributions to the values of these parameters obtained by fitting the granule cell model to granule cell data as described above. We then drew parameters randomly from these distributions for each model granule cell we generated. Parameters were the same for each input to a given granule cell, and the values of the four parameters were assumed to be independent.

## Model granule cell responses to EOD command sequences

To generate responses of model granule cells to EOD command sequences we needed to model the responses of each mossy fiber to that same sequence. We did not have a sufficiently large set of mossy fiber recordings from each class in response to EOD sequences at different rates to simply use these recordings directly as inputs to model granule cells. Instead we made simple models of how each of our previously recorded mossy fibers (whose responses only to isolated EOD commands we had recorded) would respond to EOD command sequences, based on actual responses to command sequences recorded from mossy fibers and UBCs in the present study. We considered two different models, referred to in the main text as the original model and the revised model. Medium, late and pause inputs were treated identically in the two models. Early and tonic inputs differed. For medium inputs we simply assumed that the set of spikes fired after each EOD command was the same, no matter where that command came in a sequence. This meant that spikes due to one command could overlap with spikes from subsequent commands, which we allowed, although we checked that this did not result in unrealistic firing rates of medium mossy fibers. Late inputs are characterized by a delay in firing after a command followed by a period of spiking. To model the response of a late mossy fiber to EOD command sequences we assumed that the firing delays accumulated if they overlapped. This amounts to computing the spiking response of a late mossy fiber to

an EOD command sequence by starting at the first command in the sequence and proceeding through the sequence, allowing the delay in firing following a command to prevent spikes that would otherwise have been caused by the previous command. For pause mossy fibers we estimated the length of the pause in tonic firing induced by each command. To create the response of the fiber to an EOD command sequence we drew randomly from the empirical inter-spike interval of the fiber and populated the period of the sequence with spikes. We then deleted spikes occurring within the estimated pause period after any EOD command in the sequence. This naturally gave rise to cessation of firing at high EOD command frequencies, due to accumulation of pausing. Similar responses at high command rates were observed in recorded pause mossy fibers.

## Early and tonic mossy fiber are treated differently in the two models

The key differences between the original and revised models were in the way we treated early mossy fiber inputs and tonic mossy fiber inputs. Recordings from early mossy fibers as well as mossy fiber inputs inferred from granule cell recordings showed that early mossy fibers tend to fire progressively fewer spikes per EOD command during high-frequency command sequences and that tonic mossy fibers also tend to fire at a progressively lower rate during high frequency EOD sequences. The original model does not take these new findings into account, whereas the revised model does. In the original model we assume that early mossy fibers fire the exact same burst of spikes (known from recorded responses to single EOD commands) after each command in a sequence and we create tonic mossy fiber spike trains in response to EOD command sequences by sampling from estimated inter-spike interval distributions for each recorded tonic mossy fiber. In the revised model, the fraction of spikes fired by each early mossy fiber following each EOD command, compared to the number fired after an isolated EOD command, was a function of recent EOD command history. The fraction $f$ relaxed to 1 with a characteristic timescale (80 ms) and is reduced by a factor $\alpha = 0.72$ following each EOD command:

$$\tau_f \frac{df}{dt} = 1 - f$$

and $f \rightarrow \alpha f$ after each EOD command. These parameters were chosen to approximately match the dropping observed in recorded responses of early mossy fibers to EOD command sequences. In the revised model we modified the responses of tonic mossy fibers by removing a number of spikes from the spike train based on the recent EOD command rate (computed over the last 100 ms). The decrease in tonic firing was again based on recorded tonic mossy fiber responses to EOD sequences. Tonic firing rates were decreased linearly from their maximum rate at an EOD command frequency of 10 Hz to 0.6 times their maximum rate at an EOD command frequency of 60 Hz.

## Additional granule-cell responses types

Not all granule cells from the revised model population behaved in the same way. For example, a minority of cells, specifically those receiving previously described medium mossy fiber inputs active at intermediate delays, integrate and fire more spikes at high EOD command rates. Only 2 of the 28 recorded granule cells received a medium input, consistent with the small proportion of medium inputs found previously (*Kennedy et al., 2014*). One of these cells, indeed, exhibited prominent summation and increased spiking at high rates. However, given the small proportion of medium inputs, a much larger number of actual granule cells would have to be recorded to determine whether such response types are a consistent feature of real granule cells.

## Synaptic plasticity

As in previous work we modeled the membrane potential of ELL neurons, $V(t)$, as a passive, current-based leaky unit receiving excitatory input from 20,000 model granule cells $r_i(t)$ and sensory input $s(t)$, with anti-Hebbian spike-timing dependent plasticity at granule cell-ELL neuron synapses with weights $w_i$, and EPSP kernel $E$ fit to recorded granule cell-evoked EPSPs (*Kennedy et al., 2014*). As discussed above, we adjusted the polarity of the sensory stimulus such that excitation was evoked in both E and I cells. Hence, no distinction was made in the model between E and I cells. The granule cell-ELL neuron learning rule has the form $\Delta^+ - \Delta^- L_0(t)$ where $t = t_{\text{postspike}} - t_{\text{prespike}}$ and $L_o(t)$ determines the time dependence of associative depression. Theoretical analysis has shown that

the negative images are guaranteed to be stable when $L_0 = E$, where $E$ is the EPSP from granule cells to the ELL neuron (Roberts and Bell 2000). The timescale of $E$ agrees with learning rules fit to experimental data, thus we set $L_0 = E$. We further included a regularization term as mentioned in the main text. This regularization is equivalent to a constant decay of each synaptic weight toward a baseline value that is the same for all synapses. Using this approach, the rate of change of $w_i$ is equal to $\Delta_+ \int r_i(t)dt - \Delta_- \int V(t)(E * r_i)(t)dt - \lambda(w_i - w_c)$, where the integral is over the period of the EOD command sequence being paired. The regularization constant $\lambda$ sets the time constant, $\frac{1}{\lambda}$, for the decay of synaptic weights to the baseline value $w_c$. The values of $\lambda$ and $w_c$ chosen here were selected by hand in order to match the experimental data. The model introduces these two parameters as a minimal extension of our previous model which can account for the experimental results. See the Discussion for thoughts about how these parameters might be set in the biological system.

We used $\frac{1}{\lambda} = 10s$ in the case of full regularization and $\frac{1}{\lambda} = 1000s$ for minimal regularization. The value used with full regularization was chosen to bring the overall performance of the model when generalizing as close to that found in the data without compromising cancellation at 10 Hz to the point where the model could not cancel as well as the data. The value used with minimal regularization was chosen to prevent unrealistically large weights from being learned. We chose the value of $w_c$ depending on the ELL neuron being modeled so that the mean model granule cell response scaled by $w_c$ was approximately equal to the negative of the sensory input to the ELL neuron, that is such that $w_c \langle r_i(t) \rangle \approx -s(t)$. $\Delta_+$ and $\Delta^-$ were taken from previous work, where they were fit to negative images recorded experimentally (*Kennedy et al., 2014*).

## Acknowledgments

This work was supported by grants from the NSF (1025849) to NBS and LFA and NIH (NS075023) to NBS and by the Irma T Hirschl Trust. LFA was further supported by the Simons and Gatsby Charitable Foundations and by NSF NeuroNex Award DBI-1707398.

## Additional information

### Funding

| Funder | Grant reference number | Author |
| --- | --- | --- |
| National Science Foundation | 1025849 | LF Abbott<br>Nathaniel B Sawtell |
| National Institute of Neurological Disorders and Stroke | NS075023 | Nathaniel B Sawtell |
| Irma T. Hirschl Trust | | Nathaniel B Sawtell |
| Simons Foundation | | LF Abbott |
| Gatsby Charitable Foundation | | LF Abbott |
| National Science Foundation | 1707398 | Larry F Abbott |

The funders had no role in study design, data collection and interpretation, or the decision to submit the work for publication.

### Author contributions

Conor Dempsey, Conceptualization, Formal analysis, Investigation, Methodology, Writing—original draft; LF Abbott, Conceptualization, Supervision, Writing—review and editing; Nathaniel B Sawtell, Conceptualization, Funding acquisition, Investigation, Methodology, Writing—original draft, Project administration, Writing—review and editing

### Author ORCIDs

Nathaniel B Sawtell (iD) https://orcid.org/0000-0002-1859-8026

## Ethics

Animal experimentation: All experiments performed in this study adhere to the American Physiological Society's Guiding Principles in the Care and Use of Animals and were approved by the Columbia University Institutional Animal Care and Use Committee, protocol AAAW4462.

## Decision letter and Author response

Decision letter https://doi.org/10.7554/eLife.44032.022
Author response https://doi.org/10.7554/eLife.44032.023

## Additional files

### Supplementary files

• Transparent reporting form
DOI: https://doi.org/10.7554/eLife.44032.018

### Data availability

Data and model code are available via Zenodo (doi: 10.5281/zenodo.2590782).

The following dataset was generated:

| Author(s) | Year | Dataset title | Dataset URL | Database and Identifier |
|---|---|---|---|---|
| Nathaniel Sawtell, Conor Dempsey, Larry F Abbott | 2019 | Data and model associated with "Generalization of learned responses in the mormyrid electrosensory lobe" published in eLife | http://doi.org/10.5281/zenodo.2590782 | Zenodo, 10.5281/zenodo.2590782 |

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
