## [Decision Letter]

Thank you for submitting your article "Generalization of learned responses in the mormyrid electrosensory lobe" for consideration by *eLife*. Your article has been reviewed by four peer reviewers, including Catherine Emily Carr as the Reviewing Editor and Reviewer #1, and the evaluation has been overseen by Eve Marder as the Senior Editor. The following individuals involved in review of your submission have also agreed to reveal their identity: Maurice J Chacron (Reviewer #2); Len Maler (Reviewer #3).

The reviewers have discussed the reviews with one another and the Reviewing Editor has drafted this decision to help you prepare a revised submission.

Summary:

This manuscript examines circuit-level account of generalization, using the electrosensory lobe of weakly electric mormyrid fish. Electrosensory systems show adaptive cancellation of self-generated inputs, and this paper develops a circuit-level model, with physiological recordings, to show how such systems could generalize across behavioral regimes. The mechanism involves changes in synaptic plasticity and matching of the temporal dynamics.

Essential revisions:

The reviewers do not think that extra experiments are required, given the many novel protocols and experimental results this submission already contains. The main points to be addressed are:

1) With respect to the negative image model, reviewers raised questions about a potential role for medium ganglion cells. Would a model based on the response of MGs to varying EOD rates do as well ? This could be addressed by a summary of MG cell circuitry and physiology, followed by an argument for why a first order model of negative image formation and cancellation can ignore the MGs, then future directions to bring the MGs into a more complete model.

2) There is strong spike frequency adaptation of both ampullary afferents and their ELL targets. The cited Engelmann et al., 2010, paper quantifies the receptor SFA in detail, and the authors could discuss this with respect to how EOD rate and SFA of the ampullary receptors/ELL targets might interact.

3) With respect to the model, there are ambiguities in the model description noted in the full reviews (appended). The authors should check their model against the issues raised by the reviewers.

A further point is that 'generalization' also occurs in wave-type gymnotiform fish (Bol et al., and Mejias et al.). These papers could be cited, with discussion of how cerebellar descending input could be adapted to generalizing over different EOD frequencies (mormyrid ampullary system) vs different 'noise' amplitudes (gymnotiform wave species and maybe mormyrid tuberous system).

Useful details will be found in the full reviews (appended).

*Reviewer #1:*

With respect to the negative image model and a potential role for medium ganglion cells, would a model based on the response of MGs to varying EOD rates do just as well? A serious discussion of the MGs and explanation of why they think these are not important for negative image formation and cancellation and what other role(s) they may play would be helpful.

*Reviewer #2:*

This manuscript investigates whether the cancellation signal that is internally generated by mormyrid weakly electric fish (i.e., the negative image) will cancel out EOD pulses that occur in rapid succession. The authors find that this is indeed the case and use modelling to make predictions as to the nature of the underlying mechanisms. Overall, I think that this is a very nice study. However, there are a few issues that should be dealt with:

– It is unclear how behaviorally relevant the situation considered here is to the animal. The authors do cite previous studies in the Introduction, but these appear to be dealing with object localization rather than prey capture. It should also be mentioned that the animal most likely does not exclusively use its ampullary system to locate prey but that information coming from both the passive and the active electrosense are likely combined. Finally, in looking at the authors' data, it appears that ampullary afferents and central ELL cells display strong spike frequency adaptation when EOD pulses are 15 msec apart, which might hamper their ability to detect low frequency prey stimuli. These issues need to be discussed properly.

– The model assumed by the authors uses both the "unlearned" input as well as the "learned" one. What is the physiological justification for this as I am not aware of ELL cells receiving the "unlearned" input?

– Although I like the study, it took me a few reads to fully understand the story and its implications. I suggest rewriting it such that the experimental data is presented first, then the "full" model, and the other "candidates" relegated to supplementary information. This would greatly help the reader focus on what is relevant rather than being taken through several models that do not work for various reasons.

– Finally, although the modeling is compelling, it only makes a prediction as to why negative images can "generalize" to higher frequencies, the actual nature of the mechanisms remains to be determined experimentally.

*Reviewer #3:*

This is potentially a strong manuscript but has what appears to be a serious flaw in reasoning (or in my understanding of the literature – see below). I have followed work in this system for many years but had never thought about the underlying assumptions. Mormyrids (like gymnotiform fish) have electroreceptors (ampullary) tuned to exogenous low frequency signals (e.g., respiration movements of prey). They also have tuberous receptors that are tuned to the fish's own EOD and detect changes in local EOD amplitude caused by, e.g., conductors or non-conductors. The ampullary receptors project to one map of the ELL (VLZ). This map contains E (fusiform I think) and I (large ganglion cells I think) projection neurons. These cells also have apical dendrites that are in receipt of the EOD command signal (corollary discharge) conveyed via granule cells of the overlying cerebellum (EGp).

The problem for mormyrids is that the EOD also stimulates the ampullary receptors. Their EOD triggered discharge (demonstrated decades ago by Curt Bell and colleagues) would prevent the fish from detecting prey. Curt Bell and colleagues, over many years and papers, demonstrated that a corollary discharge from EGp was able to cancel out the EOD evoked response of of VLZ neurons.

The plasticity associated with the corollary discharge input to ELL was always assayed at the convenient low EOD rates of an inactive fish. I knew that mormyrid fish can increase their EOD rates to much high levels when actively sampling their environment. I had never make the connection and question raised in this paper: do the cancellation models developed for low frequency EOD rates carry over to the high frequency EOD rate condition?

The key experiments are very nicely designed with independent control of the EOD rate and stimulus presentation. This is very nice in that pairing or not is completely under experimental control: (1) corollary discharge alone (curare: no EOD and therefore no sensory input) and experimenter stimulates corollary discharge pathway (one control); (2) sense input alone (EOD pulse mimic) in the absence of a central EOD command and therefore no corollary discharge (second control); (3) paired artificial sensory input + experimenter evoked corollary discharge (natural condition at low frequency EOD rates). Recordings are from the principle E and I cells (output cells) of VLZ.

The results, as presented in Figure 2 are striking – cancellation learned with 10 Hz paired stimulation generalizes to 40 and 60 Hz EOD rates. The careful control experiments proved very convincingly that it cancellation at the novel EOD frequencies as due to a negative image; this is a very strong extension of Curt Bell's original negative image hypothesis (for the same EOD rate for pairing and test stimulation). The nicest control is the demonstration that the response to the command stimulation alone (only corollary discharge input to E or I cells) produced a negative image. These are novel and very interesting results.

The next step was to apply a model previously developed to account for negative image formation and cancellation; this model was developed for learning and testing at the same low frequency. The response of model elements to low frequency EOD rates was extended in a simple and natural manner to the higher EOD rates used in the experiments. This model worked as expected when the test stimuli were at 10 Hz (a trivial control for the model changes). The model failed in that it over-cancelled at the higher test EOD rates. The model required training at all test frequencies and was not able to generalize thus failing to replicate the experimental data.

The original model was then changed in what appears to be minor ways (full regularization) and this did appear to improve the ability of the model to account for generalization. At this point, the authors decided that the problem lay in that they had falsely assumed the responses of the cerebellar granule cells conveying the corollary discharge signal was the same for a 10 Hz and, e.g,, a 60 Hz signal. Now, with their new protocol, they were able to record the granule cell response to the corollary discharge alone and at many different 'EOD rates'. Here a very interesting difference emerged between the data and the original model predictions: the model granule cell, but not the real granule cell, showed a pronounced temporal summation in response to the higher frequencies. A series of very nice experiments then suggested that the problem lay not in the granule cell model itself, but rather in the assumption that the mossy fibers conveying the corollary discharge responded in the same manner for 10 Hz and, e.g., 60 Hz discharge. These mossy fibers could not follow the high frequency EOD command signal – i.e., spikes dropped out. In addition, tonic mossy cell firing also was reduced.

When the authors modified the mossy fiber response component of their model, it now very nicely predicted the effect of EOD rate increases on negative image formation and cancellation. The authors explanation of what is going on is quite involved and I won't try to summarize it. It is also quite convincing.

I really could not find any flaw in the reasoning. Nonetheless, I do have one serious problem with the negative image model. To explain this, I will first summarize my possibly outdated memories of the VLZ circuitry and physiology. Any potential confusion is because there has been extensive work on both the VLZ and the mormyromast ELL zones; it is often difficult for me to determine whether conclusions are limited to ampullary vs mormyromast regions. In addition to the principle E and I cells, VLZ (and the mormyromast zones) contains medium ganglion cells (MGs). These are GABAergic interneurons with extensive apical dendrites that receive corollary discharge input from EGp granule cells. As I recall, there are two types of such MGs and they may connect differentially with the principle cell types. These MGs have narrow and broad (dendritic) spikes and the broad spikes are associated with plasticity of their corollary discharge input; in fact, I believe that Curt Bell first demonstrated parallel fiber plasticity in the VLZ MGs. The MGs provide a strong (I suspect cause of the anatomy) and plastic (Bell's data I believe) inhibitory input to principle cells. And they will be activated for both the low and high EOD rates. The authors have chosen not to incorporate these cells into their model. This seems counterintuitive to me.

I am worried that the model, though it is based on mossy fiber recordings and does a reasonable job explaining generalization, is nevertheless seriously flawed in that it omits a major source of corollary discharge input to the principle cells. Perhaps a model based on the response of MGs to varying EOD rates and including the MG to principle cell inhibitory input would do just as well as the author's model in explaining their results. The authors are certainly aware of the MGs: "…an additional prominent class of GABAergic ELL cells known as medium

ganglion cells.", but do not further discuss their potential contribution to cancellation by the principle cells.

I think the authors treatment of cerebellar Golgi cells and unipolar brush cells is OK (for a first order model). What I would like to see is a serious discussion of the MGs and explanation of why they think these are not important for negative image formation and cancellation and what other role(s) they may play. Perhaps I have gotten mixed up re VLZ and mormyromast zone data (at least some of my knowledge re comparing ampullary and morrmyromast zone cancellation comes from direct discussions with Curt Bell, Kirsty Grant and Hans Meek, and I am not even sure that all their results were even published). Basically, I want the authors to take the MGs seriously because they are a very prominent component of VLZ circuitry and must convey corollary discharge associated input to the principle cells. And then convince me that their limited model is still a reasonable first order model that will not be completely overturned once the contribution(s) of MGs are taken into account.

*Reviewer #4:*

This is a very nice study that presents novel experiments and associated modeling of negative image cancellation in the electrosensory system of mormyrid fish. The work addresses the important question of the generalizability of this cancellation from the standard low frequency sensory sampling rates used in lab settings – and by the animal at rest – to higher rates known to occur during movement. This cancellation is necessary for disentangling true external inputs from those caused by active sensing, a process that relies on a cerebellar-like corollary discharge. The authors present data that show how the animal can generalize from 10 Hz rates (training) to higher test rates.

The beauty of this study lies in particular in the intricate measurements of the granule cells and their inputs (various types of mossy fibers and unipolar brush cells), and the construction of a state-of-the-art model of a heterogeneous population of granule cells. The improved model, constructed using machine learning techniques to estimate parameters as well as the timing of spike input, reproduces the main features of the data. With their previous biophysical model thus further constrained by these new data, the improved model is shown to generalize properly, as the fish does, from the learned 10Hz sampling rate to the novel 60Hz rate. This is possible provided two ingredients are present: 1) regularization, which means that synaptic weights decay down to a value (chosen to be different for each cell), which is quicker (10s) then in their previous study (1000s) – and such an ingredient has been used for generalization in neural net studies – and there is a matching between time courses of corollary discharge and electrosensory input – although this precise statement differs in parts of the paper -see below.

The statement about the matching is confusing. In the Abstract, it is said that an approximate matching of the temporal dynamics (including EOD rate dependence) of corollary discharge and electrosensory inputs. This statement actually relies on Figure 6E and Figure 6—figure supplement 3. But in the Discussion, second paragraph of subsection “Mechanisms of generalization”, the matching is between the EOD rate dependence of electrosensory responses in ELL output cells and the rate dependence of the summed corollary discharge input that an ELL output cell receives from the granule cells. The difference is that the first case looks at the firing rate of sensory cells, and the second at the firing rate of their target cells. At the very least the statement needs to be clarified. But also I want to understand the nature of this "matching", in the sense of whether it is required to make things work or not. It seems to me that this hinges a lot on the method used to choose w_c_, i.e. the constant driving force in the regularization dynamics (subsection “Synaptic plasticity”). In other words, the ELL output cell is a function of inputs from both the electroreceptors and the granule cells, and I cannot see how to simply swap ELL and electroreceptors in the "matching" statement.

Relatedly, the choice of making w_c_ proportional to the ELL output s(t) (subsection “Synaptic plasticity”) seems to "direct" this matching, since the ELL output must depend on receptor input. The parameter w_c_ is important because it is part of both your key ingredients (regularization and matching). A clarification is also needed with regard to to precisely how w_c_ is determined: if I understand correctly, the weights w_i_ are said to evolve according to the associative learning rule (plus the regularization) which depends on the choice of w_c_; in turn w_c_ is said to depend on s(t). Yet this ELL output must depend on the w_i_ since these weights scale the 20000 granule cell inputs. So w_c_ depends on w_c_: is it self-consistently determined numerically? It would also be good to know the typical size and distribution of w_c_ values you used.

The second concern is about the regularization: your associative rule has Δ+ which drives the weights up proportionally to spike count. Yet you need w_c_ to drive them back down exponentially. It seems that one can be traded off for the other to some extent: have you checked whether Δ+ can be changed to fit the data? Was it a free parameter, or was it fixed from prior experiments?

The Discussion mentions cancellation of self-generated inputs, and thus the need for generalization in other systems, including gymnotiform weakly electric fish which have no corollary discharge. Recent experimental and modeling work (Bol et al., 2011) has shown how that system can learn to cancel many frequencies at the same time using a depressing correlation rule (not exactly the anti-hebbian rule used here) – these different frequencies are analogous in some sense to the different self-generated EOD rates the mormyrids must cancel. An elaboration of that work also showed how the experimentally observed contrast invariance of the cancellation can be achieved across frequencies (i.e. generalization to contrasts not learned, and the optimal contrast to learn at – Mejias et al., 2013). These studies require regularization, yet in that case to prevent weights from decaying to zero. An optimal rate of decay of weights was numerically predicted for the best cancellation. This sweet spot might be of a similar nature as the one of 1 𝜆λ=10 s used by trial and error by the authors (subsection “Synaptic plasticity”).

It would be important for the authors to at least make a comparative statement to this system given its close functional proximity to theirs, briefly highlighting commonalities/contrasts between corollary discharge-driven and online driven cancellation strategies. The authors' results makes interesting predictions that may apply to this system.

Finally, I would like to see a few more words about how the authors dealt with avoiding (Results paragraph three) firing-rate rectification in the ELL output cells, which they say complicated quantitative measurements of sensory cancellation. How was the polarity of the EOD mimic adjusted "to evoke excitatory responses in both E and I cells"? Which parameter does this polarity correspond to in their model? Was this parameter fitted? Heterogeneously? I can't quite wrap my head around how big or systematic a correction this was, but such a constant may influence a lot of other things, given e.g. the choice of w_c_ above.

---

## [Author Response]

Essential revisions:The reviewers do not think that extra experiments are required, given the many novel protocols and experimental results this submission already contains. The main points to be addressed are:1) With respect to the negative image model, reviewers raised questions about a potential role for medium ganglion cells. Would a model based on the response of MGs to varying EOD rates do as well ? This could be addressed by a summary of MG cell circuitry and physiology, followed by an argument for why a first order model of negative image formation and cancellation can ignore the MGs, then future directions to bring the MGs into a more complete model.

We are glad that the reviewers are interested in the function of MG cells. Though MG cells have been well-studied in some respects, e.g. in terms of their synaptic plasticity, it is important to recognize that their function is unknown, even in the context of low EOD rates studied in the past. By virtue of their anti-Hebbian plasticity, the MG cells should cancel out the effects of the EOD. However, this raises a paradox. How can MG cells aid cancellation in output cells when plasticity stops them from communicating with the rest of the ELL network? We have spent the past several years working on this problem and will shortly submit for publication a major study that offers a solution. Note, this new work focuses exclusively on the case of low rate EODs and does not address the problem of generalization. Without getting too deeply into the details, the solution involves compartmentalization of communication and learning functions within MG cells and a novel wiring scheme for which we provide experimental evidence. Essentially, this scheme allows MG cells to pass on an appropriate negative image to output cells.

Importantly, such a function is entirely consistent with the scheme for generalization proposed in the present manuscript. In light of our new work on MG cells, the question would be how can MG cells generate an appropriate negative image (to be passed onto output cells) that generalizes across rates? This brings us right back to the conclusions of the present manuscript. Namely, that regularized synaptic plasticity and an approximate matching of granule cell corollary discharge and electrosensory responses is sufficient. We would simply posit that this occurs at the level of MG cells as well as at the level of output cells. Note, there is nothing in the simplified singleneuron model of the ELL presented in the present manuscript that is inconsistent with this (i.e. the model could just as well be an MG cell).

For the purposes of the present manuscript, we do not think it is feasible or appropriate to attempt to communicate all this, given that the reasoning is based on unpublished work. However, we have added text that amplifies the important points that MG cells are likely important and that the present model is a simplification (see below). We have also removed the potentially confusing term “principal cell”.

“We also cannot rule out the importance for generalization of other circuit elements not studied here and for which we lack sufficient physiological data under conditions of different EOD rates. Our model (like all past models of the mormyrid ELL) does not distinguish between two distinct classes of ELL neurons: glutamatergic output cells versus GABAergic MG cells which inhibit output cells. MG cells occupy an analogous position in the circuitry of the mormryid ELL as Purkinje cells in the teleost cerebellum and cartwheel cells in the dorsal cochlear nucleus (Bell, 2002; Bell, Han, and Sawtell, 2008). Importantly, both MG and output cells integrate electrosensory and corollary discharge input and both exhibit anti-Hebbian plasticity (Bell, Caputi, and Grant, 1997; Bell et al., 1993; Meek et al., 1996; Mohr, Roberts, and Bell, 2003). However, it is presently unknown, even in the context of low EOD rates, how MG cells contribute to sensory cancellation and negative image formation. Our model also omits molecular layer interneurons, similar to those found in the cerebellar cortex, and does not distinguish between E- and I-type output cells. Constructing a more complete and realistic model that includes these additional features is a major focus of ongoing experimental and theoretical studies of the mormyrid ELL.”

“The model is simplified in that it does not differentiate between two distinct classes of ELL neurons: output cells and medium ganglion (MG) cells (see Discussion).”

2) There is strong spike frequency adaptation of both ampullary afferents and their ELL targets. The cited Engelmann et al., 2010, paper quantifies the receptor SFA in detail, and the authors could discuss this with respect to how EOD rate and SFA of the ampullary receptors/ELL targets might interact.

Thanks for this comment. We failed to provide a clear explanation in the main text of why responses of ampullary afferents decrease at high EOD rates. The response of ampullary afferents to a single EOD pulse consists of an increase in firing rate followed by a longer-lasting decrease below baseline. We show in Figure 6—figure supplement 3 that the response of afferent at high rates is well approximated by a linear sum of their responses to isolated pulses. This effect could indeed be related to spike frequency adaptation. This point is now discussed in the main text, see below:

“Interestingly, the sensory responses of ELL neurons to high rate trains of EOD mimics (prior to cancellation) also exhibit a decreasing profile (Figure 2A, Figure 6—figure supplement 2). A separate set of recordings from ampullary electroreceptor afferents also revealed a decreasing profile at high EOD rates (Figure 6E, Figure 6—figure supplement 3). Responses of ampullary afferents to isolated EOD pulses consist of a firing rate increase followed by a reduction below baseline and in some cases additional smaller waves of increased and decreased firing resembling a damped oscillation (Figure 6—figure supplement 3D)(Bell and Russell, 1978). Estimating the impulse response of an ampullary afferent from its average response to a single EOD mimic and then convolving this impulse response with a sequence of EOD mimics indeed yielded a reasonable approximation to the observed responses (Figure 6—figure supplement 3A, red lines). Hence the decaying profile of the sensory response to high-rate sequences of EODs as well as the inhibitory rebound at the end of such sequences are expected features of a linear system with an impulse response resembling a damped oscillation.”

Incidentally, the Engelmann paper focuses exclusively on responses to low-frequency stimuli characteristic of prey and does not characterize responses to EOD-like stimuli (extremely brief pulses ~0.2 ms in duration).

3) With respect to the model, there are ambiguities in the model description noted in the full reviews (appended). The authors should check their model against the issues raised by the reviewers.

See below.

A further point is that 'generalization' also occurs in wave-type gymnotiform fish (Bol et al., and Mejias et al.). These papers could be cited, with discussion of how cerebellar descending input could be adapted to generalizing over different EOD frequencies (mormyrid ampullary system) vs different 'noise' amplitudes (gymnotiform wave species and maybe mormyrid tuberous system).

Thanks for this, these papers are indeed relevant and are now discussed in a new paragraph in the discussion.

“The issue of generalization has been explored in the gymnotid ELL in the context of cancellation of spatially redundant electrosensory signals, such as those generated by tail movements or conspecifics (Bol et al., 2011; Mejias et al., 2013). Such cancellation is similar to that in the mormyrid ELL in that it is mediated by anti-Hebbian plasticity at synapses between granule cells and ELL neurons (Harvey-Girard, Lewis, and Maler, 2010). However, cancellation in the gymnotid ELL is driven by electrosensory feedback to granule cells rather than by corollary discharge (Bastian, Chacron, and Maler, 2004; Chacron, Maler, and Bastian, 2005). in vivo recordings from ELL neurons in gymnotids demonstrated that cancellation remains accurate over a wide range of stimulus contrasts (as might be produced by conspecifics at different distances) (Mejias et al., 2013). Modeling was used to show how learning at one contrast could generalize to higher or lower contrasts, despite numerous nonlinearities in the system. Interestingly, features of the model identified to be important for such generalization are related to those described here for the mormyrid ELL, including granule cell response properties and a slow decay in parallel fiber synaptic strength (Mejias et al., 2013), which can be considered a form of regularization. Although responses of granule cells have not yet been measured in vivo in gymnotids, several lines of evidence suggest that they are important in relation to the specificity and generalization of learning in the gymnotid ELL (Bol et al., 2011; Mejias et al., 2013).”

Useful details will be found in the full reviews (appended).

Reviewer #1:

With respect to the negative image model and a potential role for medium ganglion cells, would a model based on the response of MGs to varying EOD rates do just as well? A serious discussion of the MGs and explanation of why they think these are not important for negative image formation and cancellation and what other role(s) they may play would be helpful.

See above.

Reviewer #2:

This manuscript investigates whether the cancellation signal that is internally generated by mormyrid weakly electric fish (i.e., the negative image) will cancel out EOD pulses that occur in rapid succession. The authors find that this is indeed the case and use modelling to make predictions as to the nature of the underlying mechanisms. Overall, I think that this is a very nice study. However, there are a few issues that should be dealt with:– It is unclear how behaviorally relevant the situation considered here is to the animal. The authors do cite previous studies in the Introduction, but these appear to be dealing with object localization rather than prey capture. It should also be mentioned that the animal most likely does not exclusively use its ampullary system to locate prey but that information coming from both the passive and the active electrosense are likely combined.

Added sentence and reference to discussion:

“Behavioral studies suggest that multiple senses (including both the passive and active electrosensory systems) are used in concert to detect prey (von der Emde and Bleckmann, 1998).”

Finally, in looking at the authors' data, it appears that ampullary afferents and central ELL cells display strong spike frequency adaptation when EOD pulses are 15 msec apart, which might hamper their ability to detect low frequency prey stimuli. These issues need to be discussed properly.

See above.

– The model assumed by the authors uses both the "unlearned" input as well as the "learned" one. What is the physiological justification for this as I am not aware of ELL cells receiving the "unlearned" input?

We are referring to the peripheral electrosensory input to ELL neurons. Text has been added for clarification. This input is known anatomically and is central to ELL function.

“This corresponds anatomically to the input onto the basilar dendrites of ELL neurons from interneurons in the deep layers of ELL receiving somatotopic input from ampullary electroreceptor afferents (Meek, Grant, and Bell, 1999)”

– Although I like the study, it took me a few reads to fully understand the story and its implications. I suggest rewriting it such that the experimental data is presented first, then the "full" model, and the other "candidates" relegated to supplementary information. This would greatly help the reader focus on what is relevant rather than being taken through several models that do not work for various reasons.

We appreciate the reviewer’s patience and agree that model failures should not be the focus. We tried numerous different versions of the manuscript, including some in which we separated the experimental and theoretical results. In the end, we felt that integrating the experiments and the models, as we have tried to do, was best. In the revised manuscript we have added some material to the Introduction to better set up the problem and preview our main findings (per the suggestion of reviewer 2 below). We think this will help readability.

– Finally, although the modeling is compelling, it only makes a prediction as to why negative images can "generalize" to higher frequencies, the actual nature of the mechanisms remains to be determined experimentally.

We agree that the model has not been proven. On the other hand, the model does provide a functional interpretation for previously unknown and unexpected features of mossy fiber and granule cell corollary discharge responses reported here for the first time. For example, the observation from past work that mossy fibers fire identical bursts of ~8 precisely timed spike in response to each command seems odd. Why are 8 spikes needed to signal the timing of the EOD when a single spike would seemingly suffice? Here we show experimentally that the mossy fiber bursts change with EOD rate and the model provides a non-trivial, mechanistic explanation for how information encoded in the EOD command rate-dependence of the mossy fibers could be used to support generalization of learning.

“So-called early mossy fibers are known from previous studies to fire a highly-stereotyped burst of action potentials following each EOD command (Bell, Grant, and Serrier, 1992). We found that the number of spikes in such bursts declines progressively with increases in the command rate. The multiple spikes in the burst seem redundant in the context of isolated EODs. Why would multiple spikes be needed to signal the time of occurrence of an EOD command? The present work suggests that rate-dependent grading of such bursts conveys information that is important for generalization. “

Reviewer #3:This is potentially a strong manuscript but has what appears to be a serious flaw in reasoning (or in my understanding of the literature – see below). I have followed work in this system for many years but had never thought about the underlying assumptions. Mormyrids (like gymnotiform fish) have electroreceptors (ampullary) tuned to exogenous low frequency signals (e.g., respiration movements of prey). They also have tuberous receptors that are tuned to the fish's own EOD and detect changes in local EOD amplitude caused by, e.g., conductors or non-conductors. The ampullary receptors project to one map of the ELL (VLZ). This map contains E (fusiform I think) and I (large ganglion cells I think) projection neurons. These cells also have apical dendrites that are in receipt of the EOD command signal (corollary discharge) conveyed via granule cells of the overlying cerebellum (EGp).[…]I think the authors treatment of cerebellar Golgi cells and unipolar brush cells is OK (for a first order model). What I would like to see is a serious discussion of the MGs and explanation of why they think these are not important for negative image formation and cancellation and what other role(s) they may play. Perhaps I have gotten mixed up re VLZ and mormyromast zone data (at least some of my knowledge re comparing ampullary and morrmyromast zone cancellation comes from direct discussions with Curt Bell, Kirsty Grant and Hans Meek, and I am not even sure that all their results were even published). Basically, I want the authors to take the MGs seriously because they are a very prominent component of VLZ circuitry and must convey corollary discharge associated input to the principle cells. And then convince me that their limited model is still a reasonable first order model that will not be completely overturned once the contribution(s) of MGs are taken into account.

Agreed. See above.

Reviewer #4:

This is a very nice study that presents novel experiments and associated modeling of negative image cancellation in the electrosensory system of mormyrid fish. The work addresses the important question of the generalizability of this cancellation from the standard low frequency sensory sampling rates used in lab settings – and by the animal at rest – to higher rates known to occur during movement. This cancellation is necessary for disentangling true external inputs from those caused by active sensing, a process that relies on a cerebellar-like corollary discharge. The authors present data that show how the animal can generalize from 10 Hz rates (training) to higher test rates.[…]The statement about the matching is confusing. In the Abstract, it is said that an approximate matching of the temporal dynamics (including EOD rate dependence) of corollary discharge and electrosensory inputs. This statement actually relies on Figure 6E and Figure 6—figure supplement 3. But in the Discussion, second paragraph of subsection “Mechanisms of generalization”, the matching is between the EOD rate dependence of electrosensory responses in ELL output cells and the rate dependence of the summed corollary discharge input that an ELL output cell receives from the granule cells. The difference is that the first case looks at the firing rate of sensory cells, and the second at the firing rate of their target cells. At the very least the statement needs to be clarified. But also I want to understand the nature of this "matching", in the sense of whether it is required to make things work or not. It seems to me that this hinges a lot on the method used to choose w_c_, i.e. the constant driving force in the regularization dynamics (subsection “Synaptic plasticity”). In other words, the ELL output cell is a function of inputs from both the electroreceptors and the granule cells, and I cannot see how to simply swap ELL and electroreceptors in the "matching" statement.

The matching is between the average granule cell response and the electrosensory inputs to ELL output cells. The electrosensory response of the output cells is very similar to the electrosensory response of the afferent fibers providing input to those cells prior to the learning of negative images, which is the reason for the confusing statements where the electrosensory response of output cells is said to match the average granule cell response. We have changed the statement in the Discussion to clarify this:

“The second feature we identified as important for generalization is an approximate matching between the EOD rate dependence of electrosensory inputs to ELL output cells and the rate dependence of the average corollary discharge response in granule cells.”

Relatedly, the choice of making w_c_ proportional to the ELL output s(t) (subsection “Synaptic plasticity”) seems to "direct" this matching, since the ELL output must depend on receptor input. The parameter w_c_ is important because it is part of both your key ingredients (regularization and matching). A clarification is also needed with regard to to precisely how w_c_ is determined: if I understand correctly, the weights w_i_ are said to evolve according to the associative learning rule (plus the regularization) which depends on the choice of w_c_; in turn w_c_ is said to depend on s(t). Yet this ELL output must depend on the w_i_ since these weights scale the 20000 granule cell inputs. So w_c_ depends on w_c_: is it self-consistently determined numerically? It would also be good to know the typical size and distribution of w_c_ values you used.

This is a simple confusion: s(t) is the “sensory only response”, not the summed response to corollary discharge and sensory input. We have added the following clarifying statement.

“We chose the value of 𝑤_𝑐_ depending on the ELL neuron being modeled so that the mean model granule cell response scaled by 𝑤_𝑐_ was approximately equal to the negative of the sensory input to the ELL neuron, that is such that 𝑤_𝑐_⟨𝑟_𝑖_(𝑡)⟩ ≈ −𝑠(𝑡). “

The second concern is about the regularization: your associative rule has Δ+ which drives the weights up proportionally to spike count. Yet you need w_c_ to drive them back down exponentially. It seems that one can be traded off for the other to some extent: have you checked whether Δ+ can be changed to fit the data? Was it a free parameter, or was it fixed from prior experiments?

Δ+ was estimated from previous in vitro experiments, Bell, 1997 and Han, 2000. Regularization and Δ+/Δ- cannot be traded off against one another. The ratio of Δ+/Δ- sets the steady-state firing rate of the cell, and the scale of the two parameters sets the rate of learning. Regularization is a decay toward w_c_ in this model, which can involve both potentiation and depression toward w_c_.

The Discussion mentions cancellation of self-generated inputs, and thus the need for generalization in other systems, including gymnotiform weakly electric fish which have no corollary discharge. Recent experimental and modeling work (Bol et al., 2011) has shown how that system can learn to cancel many frequencies at the same time using a depressing correlation rule (not exactly the anti-hebbian rule used here) – these different frequencies are analogous in some sense to the different self-generated EOD rates the mormyrids must cancel. An elaboration of that work also showed how the experimentally observed contrast invariance of the cancellation can be achieved across frequencies (i.e. generalization to contrasts not learned, and the optimal contrast to learn at – Mejias et al., 2013). These studies require regularization, yet in that case to prevent weights from decaying to zero. An optimal rate of decay of weights was numerically predicted for the best cancellation. This sweet spot might be of a similar nature as the one of 1 𝜆=10 s used by trial and error by the authors (subsection “Synaptic plasticity”).It would be important for the authors to at least make a comparative statement to this system given its close functional proximity to theirs, briefly highlighting commonalities/contrasts between corollary discharge-driven and online driven cancellation strategies. The authors' results makes interesting predictions that may apply to this system.

See above. Thanks for this, these papers are indeed relevant and are now discussed in a new paragraph in the discussion.

Finally, I would like to see a few more words about how the authors dealt with avoiding (Results paragraph three) firing-rate rectification in the ELL output cells, which they say complicated quantitative measurements of sensory cancellation. How was the polarity of the EOD mimic adjusted "to evoke excitatory responses in both E and I cells"? Which parameter does this polarity correspond to in their model? Was this parameter fitted? Heterogeneously? I can't quite wrap my head around how big or systematic a correction this was, but such a constant may influence a lot of other things, given e.g. the choice of w_c_ above.

Text has been added to the experimental and modeling sections of the Materials and methods section:

“In I cells, the EOD mimic often drove the firing rate to zero, making it difficult to quantify cancellation. To avoid this firing rate rectification, we reversed the EOD mimic polarity when recording from I cells, such that they responded with excitation instead of inhibition. This response reversal is due to known properties of ampullary electroreceptor afferents, which increase (or decrease) firing above (or below) their baseline rate for stimuli that make the pore of the receptor positive (or negative) with respect to the basal face within the body. Past studies have commonly used this approach to demonstrate the specificity of negative image formation in the VLZ by performing multiple pairing in the same neuron using opposite stimulus polarities (Bell, 1981, 1982; Enikolopov, Abbott, and Sawtell, 2018). Negative images are invariably observed for both stimulus polarities in such experiments, i.e. responses to the corollary discharge alone after pairing are opposite in sign to the response to the stimulus during pairing. Systematic differences between negative images and cancellation in E versus I cells or for mimics of opposite polarities have never been noted, justifying this approach to avoid rectification.”

“As discussed above, we adjusted the polarity of the sensory stimulus such that excitation was evoked in both E and I cells. Hence, no distinction was made in the model between E and I cells.”